# EigenScore: OOD Detection using Posterior Covariance in Diffusion Models

**Shirin Shoushtari**
Washington University in St. Louis
s.shirin@wustl.edu

**Yi Wang**
Washington University in St. Louis
y.wang1@wustl.edu

**Xiao Shi**
University of Wisconsin–Madison
xiao.shi@wisc.edu

**M. Salman Asif**
University of California, Riverside
sasif@ucr.edu

**Ulugbek S. Kamilov**
University of Wisconsin–Madison
kamilov@wisc.edu

## Abstract

Out-of-distribution (OOD) detection is critical for the safe deployment of machine learning systems in safety-sensitive domains. Diffusion models have recently emerged as powerful generative models, capable of capturing complex data distributions through iterative denoising. Building on this progress, recent work has explored their potential for OOD detection. We propose *EigenScore*, a new OOD detection method that leverages the eigenvalue spectrum of the posterior covariance induced by a diffusion model. We argue that posterior covariance provides a consistent signal of distribution shift, leading to larger trace and leading eigenvalues on OOD inputs, yielding a clear spectral signature. We further provide analysis explicitly linking posterior covariance to distribution mismatch, establishing it as a reliable signal for OOD detection. To ensure tractability, we adopt a Jacobian-free subspace iteration method to estimate the leading eigenvalues using only forward evaluations of the denoiser. Empirically, EigenScore achieves state-of-the-art performance, with up to 2% AUROC improvement over the best baseline. Notably, it remains robust in near-OOD settings such as CIFAR-10 vs CIFAR-100, where existing diffusion-based methods often fail.

## 1 Introduction

Most machine learning systems assume that test data matches the training distribution, but distribution shift or out-of-distribution (OOD) data can severely degrade performance in safety-critical domains such as medical imaging and autonomous driving (Yang et al., 2024). OOD inputs may stem from sensor noise, semantic differences, or acquisition changes, leading to unreliable predictions (Zhang et al., 2023). To address this, many OOD detection methods have been proposed, ranging from supervised approaches that require labeled OOD data to unsupervised approaches that rely only on in-distribution (InD) training data (Graham et al., 2023).

Existing OOD detection methods can be broadly categorized into four families: (i) **uncertainty-based** methods, which rely on signals such as softmax confidence (Hendrycks & Gimpel, 2017), ensemble variance (Choi et al., 2018), or Bayesian inference (Wang & Aitchison, 2021; Charpentier et al., 2020) to identify anomalous inputs; (ii) **distance-based** methods (Regmi et al., 2024), which compare test embeddings to in-distribution features, commonly via Mahalanobis distance (Colombo et al., 2022; Lee et al., 2018); (iii) **density-based** methods (Huang et al., 2022), including flow- and energy-based models (Kumar et al., 2023; Liu et al., 2020), which attempt to estimate likelihoods but have been shown to assign spuriously high likelihoods to OOD data (Nalisnick et al., 2019a); and (iv) **representation-learning** methods (Wang et al., 2022), including self-supervised and contrastive

techniques (Seifi et al., 2024; Hendrycks et al., 2019; Tack et al., 2020), which improve robustness by explicitly shaping feature spaces (Also see reviews in Koh & et. al (2021); Salehi et al. (2022)).

Diffusion models (DMs) (Ho et al., 2020; Song et al., 2020) have emerged as state-of-the-art generative models, achieving high-quality samples across diverse domains. Their success has spurred both architectural advances (Vahdat et al., 2021; Dhariwal & Nichol, 2021; Rombach et al., 2022; Karras et al., 2022) and applications beyond generation, such as imaging inverse problems and medical tasks (Chung et al., 2023; Adib et al., 2023) (see also recent reviews (Daras et al., 2024; Croitoru et al., 2023; Li et al., 2023; Kazerouni et al., 2023)). Crucially, diffusion models are especially relevant for OOD detection because their iterative denoising process does not simply produce samples, but also provides access to score functions that explicitly characterize the data distribution. Early work exploited this property through likelihood- or reconstruction-based scores (Graham et al., 2023; Gao et al., 2023; He et al., 2025). More recent studies have explored structural aspects of the diffusion trajectory, such as score geometry and intermediate representations (Heng et al., 2024; Graham et al., 2023; Liu et al., 2023; Choi et al., 2023). These developments highlight both the promise of diffusion-based OOD methods and the need for principled approaches that move beyond heuristic scoring rules.

Building on recent work in diffusion-based OOD detection, we introduce **EigenScore**, an unsupervised, feature-based framework for identifying distribution shift. Unlike reconstruction-based methods that measure input–output similarity (Graham et al., 2023) or trajectory-based methods that analyze diffusion-path geometry (Heng et al., 2024), EigenScore leverages the covariance structure of the denoising process to capture uncertainty signals. In particular, DDPM-OOD (Graham et al., 2023) detects distribution shift by measuring reconstruction fidelity, using perceptual or pixel-space discrepancies between the input and its denoised reconstruction as an OOD score. While effective in some regimes, this strategy implicitly assumes that distribution shift manifests primarily through degraded reconstruction quality. In contrast, EigenScore does not depend on reconstruction error magnitude or perceptual similarity; instead, it probes the posterior uncertainty of the denoiser by analyzing the eigenvalue spectrum of the posterior covariance. As shown in Section 3.1, even when OOD inputs admit visually plausible reconstructions—as is common in near-OOD settings—the posterior covariance inflates in structured directions, yielding a consistent and theoretically grounded signal of distribution shift.

Recent work by Kamkari et al. (2024) provides a geometric explanation of the likelihood-based OOD detection paradox by analyzing the local geometry of generative models. Their approach studies the Jacobian of the generative mapping and shows that likelihood scores conflate radial distance to the data manifold with tangential volume effects, leading to misleading OOD behavior. While their method also involves spectral quantities through singular values of a Jacobian-related matrix, the motivation and signal differ fundamentally from ours. EigenScore does not operate on likelihoods or generative mappings; instead, it analyzes the posterior covariance of the diffusion denoiser, capturing predictive uncertainty rather than volume distortion. As a result, EigenScore yields a stable uncertainty-based signal even in regimes where likelihood geometry is known to fail, such as near-OOD settings.

By explicitly linking posterior covariance, estimated from the denoiser's Jacobian, to distribution mismatch, EigenScore provides an interpretable uncertainty-based signal while remaining practical at scale through a Jacobian-free eigenvalue estimation algorithm. Our theoretical analysis and empirical results demonstrate that applying an in-distribution diffusion model to OOD samples leads to systematic inflation of posterior covariance, enabling stable and discriminative OOD detection. Our contributions are threefold:

- We introduce **EigenScore**, an unsupervised, feature-based framework for OOD detection in diffusion models. EigenScore leverages the posterior covariance of the denoising process to characterize distribution shift.

- We provide supporting analysis establishing a direct connection between denoising uncertainty from posterior covariance and distribution mismatch, thereby explaining why EigenScore reliably separates InD from OOD samples.

- We conduct extensive experiments on standard OOD benchmarks (CIFAR-10 (C10), CIFAR-100 (C100), SVHN, CelebA, TinyImageNet), showing that EigenScore achieves average state-of-the-art performance and remains notably robust in challenging near-OOD scenarios.

## 2 BACKGROUND

**Diffusion Models.** Diffusion models learn to generate samples by simulating a gradual denoising process. During training, a clean sample $\boldsymbol{x} \sim p(\boldsymbol{x})$ is perturbed by Gaussian noise across timesteps $t = 1, \cdots, T$, producing noisy states $\boldsymbol{x}_t$ through the forward Markov chain $p(\boldsymbol{x}_t|\boldsymbol{x}) = \mathcal{N}(\boldsymbol{x}, \sigma_t^2 \boldsymbol{I})$, which allows for direct sampling via $\boldsymbol{x}_t = \boldsymbol{x} + \boldsymbol{z}$, where $\boldsymbol{z} \sim \mathcal{N}(0, \sigma_t^2 \boldsymbol{I})$.

The reverse process is approximated by a denoising network $\mathsf{D}_\theta(\boldsymbol{x}_t, t)$, trained to predict either the clean signal or the injected noise. A standard training objective is mean squared error (MSE):

$$\mathcal{L}_{\mathrm{MSE}}(\mathsf{D}_\theta) = \mathbb{E}_{\boldsymbol{x},\boldsymbol{x}_t,t} \left[ \left\| \boldsymbol{x} - \mathsf{D}_\theta(\boldsymbol{x}_t, t) \right\|_2^2 \right]. \tag{1}$$

Once trained, the model generates new samples by iteratively denoising from Gaussian noise at $t = T$ back to $t = 0$. Importantly, Tweedie's formula (Robbins, 1956; Miyasawa, 1961) connects Gaussian denoising with score estimation, linking the posterior mean to the gradient of the log-density as

$$\mathsf{D}_p(\boldsymbol{x}_t) = \mathbb{E}_p[\boldsymbol{x}|\boldsymbol{x}_t] = \boldsymbol{x}_t + \sigma_t^2 \nabla \log p(\boldsymbol{x}_t), \tag{2}$$

where $\mathsf{D}_p(\boldsymbol{x}_t)$ denotes an MMSE estimator trained on samples from distribution $p$. Here, the gradient is with respect to $\boldsymbol{x}_t$, and $p(\boldsymbol{x}_t)$ denotes the marginal distribution noisy image

$$p(\boldsymbol{x}_t) = \int p(\boldsymbol{x}_t|\boldsymbol{x})p(\boldsymbol{x})\mathrm{d}\boldsymbol{x} = \int G_{\sigma_t}(\boldsymbol{x}_t - \boldsymbol{x})p(\boldsymbol{x})\mathrm{d}\boldsymbol{x}, \tag{3}$$

where $G_{\sigma_t}$ denotes the Gaussian density function with standard deviation $\sigma_t \geq 0$ (Vincent, 2011; Raphan & Simoncelli, 2011). This relationship implies that denoising does more than produce samples—it provides access to score functions and posterior statistics of the underlying distribution. In the context of OOD detection, this observation motivates our use of the denoiser's covariance structure as a principled signal of distribution shift. Diffusion models admit several formulations (e.g., variance-preserving, variance-exploding, and SDE-based), but all share the key property of learning the score function $\nabla_{\boldsymbol{x}_t} \log p(\boldsymbol{x}_t)$ to guide denoising (Ho et al., 2020; Song et al., 2020; Song & Ermon, 2019; Yang et al., 2023).

**Unsupervised OOD detection.** Unsupervised OOD detection aims to determine whether a given sample $\boldsymbol{x}$ originates from the same distribution as the training data, using only unlabeled InD samples $\boldsymbol{x}_1, \cdots, \boldsymbol{x}_n \sim p(\boldsymbol{x})$. The goal is to learn a detector that assigns an OOD score to each input, where higher scores indicate a greater likelihood that $\boldsymbol{x}$ was drawn from a different distribution, such as the OOD density $q(\boldsymbol{x})$ (Graham et al., 2023; Heng et al., 2024).

**Likelihood-based methods.** These methods use generative models including VAEs, flows, diffusion models to estimate sample likelihoods, under the assumption that OOD data should receive lower likelihoods (Salimans et al., 2017; Kingma & Dhariwal, 2018; Morningstar et al., 2021; Ding et al., 2025). However, it has been shown that generative models often assign high likelihoods to OOD inputs (Choi et al., 2018; Nalisnick et al., 2019a; Kirichenko et al., 2020). To mitigate this, refined scores have been proposed, including likelihood ratios (Ren et al., 2019), compression corrections (Serrà et al., 2019), WAIC ensembles (Choi et al., 2018), and typicality tests (Nalisnick et al., 2019b). Diffusion-based variants further extend this idea by analyzing statistics across the denoising trajectory (Heng et al., 2024; Livernoche et al., 2024).

**Reconstruction-based methods.** Another line of work assumes that InD samples reconstruct well, whereas OOD samples do not. Early examples include autoencoders (Zhou & Paffenroth, 2017) and GANs (Schlegl et al., 2019). More recently, diffusion models have been exploited for their strong reconstruction fidelity, leading to perceptual quality scores (Graham et al., 2023), projection regret (Choi et al., 2023), and masked inpainting like LMD (Liu et al., 2023). OOD sample detection can alse be via subspace reconstruction of features or gradients, using PCA (Guan et al., 2023), kernel PCA (Fang et al., 2024), or gradient projections such as GradOrth (Behpour et al., 2023).

**Feature-based methods.** These approaches distinguish InD from OOD by leveraging learned representations, such as Mahalanobis distance in latent space (Denouden et al., 2018), unsupervised contrastive features (Hendrycks et al., 2019; Bergman & Hoshen, 2020; Tack et al., 2020), or encoder features from invertible models (Ahmadian & Lindsten, 2021). Pretrained feature extractors have also proven effective (Xiao et al., 2020).

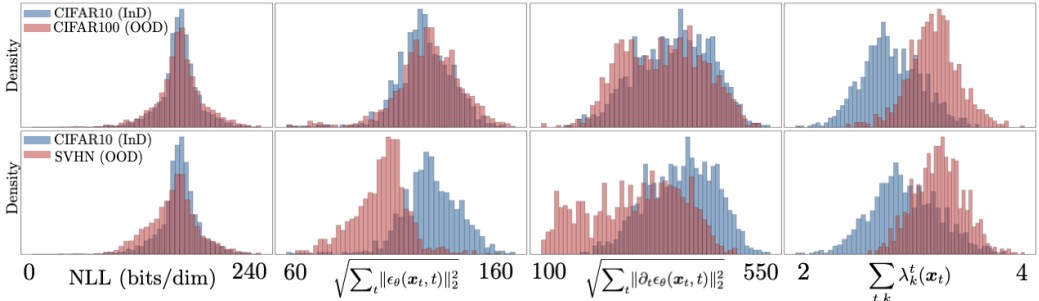

Figure 1: *We compare negative log-likelihood (NLL), score norm $\sqrt{\sum_t \|\epsilon_\theta(\boldsymbol{x}_t, t)\|_2^2}$, score derivative norm $\sqrt{\sum_t \|\partial_t \epsilon_\theta(\boldsymbol{x}_t, t)\|_2^2}$, and the eigenvalue sum (ours) $\sum_{t,k} \lambda_k^t(\boldsymbol{x}_t)$ as OOD detection statistics. **Top row: near OOD task** for C10 (InD) vs. C100, NLL and score-based metrics fail to separate distributions, showing substantial overlap. **Bottom row:** for C10 (InD) vs. SVHN (OOD), the ordering of metrics inverts—score and derivative norms assign* lower *values to OOD than InD, making thresholds unreliable. In both settings, our eigenvalue-based metric achieves clear separation and consistently assigns higher scores to OOD samples.*

Complementing these approaches, we introduce a new perspective based on posterior covariance in diffusion models, which provides a principled feature for quantifying distribution shift.

## 3 DIFFUSION MODEL FOR OUT-OF-DISTRIBUTION DETECTION

EigenScore is a novel OOD detection method that exploits covariance structure of the denoising diffusion process. Our key insight is that when a diffusion model trained on InD data is applied to OOD inputs, the variance of its denoising predictions inflates, leaving a characteristic signature in the eigenvalue spectrum of the score Jacobian.

To motivate EigenScore, we first revisit why commonly used diffusion-based OOD metrics—likelihood, score norm, and score derivatives—are unreliable (Sec. 3.1). We then show, both theoretically and empirically, that posterior covariance offers a consistent marker of distribution shift (Sec. 3.2), before formalizing EigenScore and its efficient computation (Sec. 3.3).

### 3.1 WHY LIKELIHOOD AND SCORE DYNAMICS ARE INSUFFICIENT

Since diffusion models are trained via a variational lower bound (ELBO), likelihood-based scores such as negative log-likelihood (NLL) are natural candidates for OOD detection. However, likelihood does not necessarily align with semantic structure: diffusion models often emphasize low-level statistics while ignoring higher-level semantics, making NLL unreliable (Nalisnick et al., 2019b; Serrà et al., 2019). Empirically, NLL can even assign higher likelihoods to OOD samples than to InD ones (Heng et al., 2024). As shown in Fig. 1, NLL is not a reliable metric for separating InD from OOD samples.

Beyond likelihood, diffusion-based OOD metrics have also used the score function $\epsilon_\theta(\boldsymbol{x}_t, t)$ and its temporal derivative $\partial_t \epsilon_\theta(\boldsymbol{x}_t, t)$ as statistics (Heng et al., 2024). Their norms provide some empirical separation, but they remain unstable. In near-OOD settings (C10 vs. C100), the distributions overlap substantially (Fig.1, top row). In some settings (C10 vs. SVHN), the ordering can invert, with OOD samples receiving lower scores than InD (Fig.1, bottom row). These limitations motivate shifting from scalarized scores toward a covariance-based perspective, where the structure of denoiser variability itself provides a more principled signal of distribution shift

### 3.2 UNCERTAINTY AS A SIGNAL OF DISTRIBUTION SHIFT

We formalize why denoising uncertainty yields a principled OOD signal. Let $p(\boldsymbol{x})$ denote InD and $q(\boldsymbol{x})$ an OOD distribution. Under Gaussian corruption with variance $\sigma_t^2$, the KL divergence admits the score-based representation (Song et al., 2021; Kadkhodaie et al., 2024; Shoushtari et al., 2025;

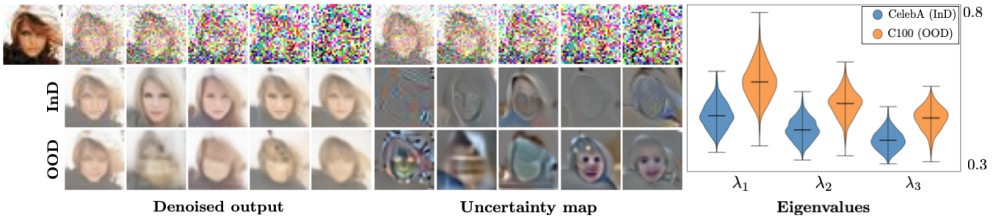

*Figure 2: Denoised outputs (left), corresponding uncertainty maps (first principle component) (middle), and violin plots of the three largest eigenvalues for CelebA dataset (right). Top: clean CelebA image and its noisy variants for varying t. Middle: InD model (trained on CelebA) applied to CelebA inputs. Bottom: OOD model (trained on C100) applied to the same inputs. InD models yield sharp reconstructions and localized uncertainty with smaller leading eigenvalues, whereas OOD models produce blurrier outputs, diffuse uncertainty, and inflated eigenvalues—highlighting the eigenvalue spectrum as an indicator of distribution shift.*

---

**Algorithm 1** OOD Detection with EigenScore — Train/Validation (left) and Test (right)

| | |
|---|---|
| **Train/Validation:** Select time-steps, aggregation, and compute Z-score stats | **Test:** Compute EigenScore |
| **Require:** Trained DM $D_p$, train set $\mathcal{X}_{\text{train}}$, $K$ number of eigenvalues, $I$ number of repetition, $L_{\text{train}} = [\,]$ | **Require:** Trained DM $D_p$, test set $\mathcal{X}_{\text{test}}$, number of eigenvalues $K$, number of repetitions $I$, $(T^*, agg^*, \{\mu_t, \sigma_t\}_{t=1}^{T^*})$, $L_{\text{test}} = [\,]$ |
| 1: **for** $\boldsymbol{x} \in \mathcal{X}_{\text{train}}$ **do** | 1: **for** $\boldsymbol{x} \in \mathcal{X}_{\text{test}}$ **do** |
| 2:     Compute $\mathbf{M}(\boldsymbol{x})$ via Eq. (10) | 2:     Compute $\mathbf{M}(\boldsymbol{x})$ using $T^*$ and $agg^*$ |
| 3:     Append $\mathbf{M}(\boldsymbol{x})$ to $L_{\text{train}}$ | 3:     $z_t(\boldsymbol{x}) = \frac{\overline{m}_t(\boldsymbol{x}) - \mu_t}{\sigma_t}$ for $t = 1, \dots, T^*$ |
| 4: **end for** | 4:     $S_\theta(\boldsymbol{x}) = \sum_{t=1}^{T^*} z_t(\boldsymbol{x})$ |
| 5: Compute $\mu_t, \sigma_t$ across $L_{\text{train}}$ | 5:     Append $S_\theta(\boldsymbol{x})$ to $L_{\text{test}}$ |
| 6: Use validation set to tune $T$ and aggregation method (mean/median/none) | 6: **end for** |
| 7: **return** $(T^*, agg^*, \{\mu_t, \sigma_t\}_{t=1}^{T^*})$ | 7: **return** $L_{\text{test}}$     ▷ OOD scores for all test samples |

---

Heng et al., 2024)

$$\mathrm{D_{KL}}(p \parallel q) = \int_0^T \mathbb{E}_{\boldsymbol{x}, \boldsymbol{x}_t} \left[ \left\| \nabla \log p(\boldsymbol{x}_t) - \nabla \log q(\boldsymbol{x}_t) \right\|_2^2 \right] \sigma_t \, dt. \tag{4}$$

where $\boldsymbol{x}_t$ is generated by the forward diffusion applied to $\boldsymbol{x}$. Building on prior analyses of KL divergence in diffusion processes (Shoushtari et al., 2025; Heng et al., 2024), we restate the divergence in terms of denoising error (a derivation is given in App. A.1 for completeness).

**Proposition 1.** *Let $p_t$ and $q_t$ denote the noisy marginals of InD and OOD distributions generated by the forward diffusion process (Eq. (3)). For MMSE denoisers $D_p(\boldsymbol{x}_t) = \mathbb{E}_p[\boldsymbol{x}|\boldsymbol{x}_t]$ and $D_q(\boldsymbol{x}_t) = \mathbb{E}_q[\boldsymbol{x}|\boldsymbol{x}_t]$,*

$$\mathrm{D_{KL}}(p \parallel q) = \int_0^T \left[ MSE(D_q, t) - MSE(D_p, t) \right] \sigma_t^{-3} dt$$

*where $MSE(D_p, t) = \mathbb{E}\left[ \|\boldsymbol{x} - D_p(\boldsymbol{x}_t)\|_2^2 \right]$ at noise level $t$.*

This proposition, adapted from earlier derivations, shows that KL divergence—and thus distribution shift—can be viewed as the accumulation of *excess denoising error* incurred when contrasting the optimal MMSE denoiser under $q$ with that under $p$. Thus, we have $\mathrm{MSE}(D_p; q, t) \geq \mathrm{MSE}(D_q; q, t)$ for each $t$ and the denoising error of a single InD denoiser $D_p$ is *larger in expectation* on OOD inputs, yielding a practical detection signal without access to $q$.

For the MMSE denoiser $D_p$, the mean-squared error admits a law-of-total-variance decomposition (proof in App. A.3):

$$\mathrm{MSE}(D_p, t) = \mathbb{E}\left[ \|\boldsymbol{x} - D_p(\boldsymbol{x}_t)\|_2^2 \right] = \mathbb{E}_{\boldsymbol{x}_t} \left[ \mathrm{tr}\left( \mathrm{Cov}_p[\boldsymbol{x} \mid \boldsymbol{x}_t] \right) \right]. \tag{5}$$

Thus, denoising error equals the *total posterior variance*—the trace of the conditional covariance—averaged over noisy observations at noise level $t$. Intuitively, $\mathrm{Cov}_p[\boldsymbol{x} \mid \boldsymbol{x}_t]$ quantifies the

spread of plausible clean signals consistent with $\boldsymbol{x}_t$. When $\boldsymbol{x}_t$ is informative (e.g., InD or low noise), this spread is small and the MSE remains low; when $\boldsymbol{x}_t$ lies off-manifold, the spread inflates and the MSE increases. In this sense, denoising error corresponds directly to the model's predictive uncertainty, providing a principled basis for OOD detection and motivating our focus on the *spectral structure* of posterior covariance in Section 3.3.

While Eq.4 relates distribution shift to differences in score norms, such metrics do not preserve ordering: depending on the dataset and noise scale, OOD samples may appear either larger or smaller than InD ones. By contrast, the MSE-based formulation in Prop.1 guarantees that OOD denoising error is systematically larger in expectation than InD error. This ensures a consistent separation with preserved ordering, whereas score-norm methods such as DiffPath (Heng et al., 2024) capture only relative differences without indicating direction.

## 3.3 EIGENVALUE-BASED UNCERTAINTY ESTIMATION

Section 3.2 established that denoising error equals the total posterior variance and that this variance inflates under distribution shift. We now make this connection operational by expressing posterior covariance through the Jacobian of the MMSE denoiser and analyzing its eigen-spectrum. For the MMSE denoiser $\mathsf{D}_p(\boldsymbol{x}_t) = \mathbb{E}[\boldsymbol{x}|\boldsymbol{x}_t]$, the posterior covariance admits the Miyasawa identity (Miyasawa, 1961):

$$\mathrm{Cov}_p[\boldsymbol{x}|\boldsymbol{x}_t] = \sigma_t^2 \left( \boldsymbol{I} + \sigma_t^2 \nabla^2 \log p(\boldsymbol{x}_t) \right) = \sigma_t^2 \nabla \mathsf{D}_p(\boldsymbol{x}_t), \tag{6}$$

so that

$$\mathrm{MSE}(\mathsf{D}_p, t) = \mathbb{E}_{\boldsymbol{x}_t} \left[ \mathrm{tr}\left( \mathrm{Cov}_p[\boldsymbol{x}|\boldsymbol{x}_t] \right) \right] = \sigma_t^2 \mathbb{E}_{\boldsymbol{x}_t} \left[ \mathrm{tr}\left( \nabla \mathsf{D}_p(\boldsymbol{x}_t) \right) \right], \tag{7}$$

Thus, the Jacobian trace measures how much posterior uncertainty remains after observing $\boldsymbol{x}_t$: small traces indicate concentrated beliefs (InD), while large traces signal inflated uncertainty (OOD). Derivations are provided in Appendix (A.2).

Writing $\boldsymbol{\Sigma}_t(\boldsymbol{x}_t) := \mathrm{Cov}_p[\boldsymbol{x}|\boldsymbol{x}_t] = \sigma_t^2 \nabla \mathsf{D}_p(\boldsymbol{x}_t)$, the covariance is symmetric positive semi-definite and admits the eigen-decomposition $\boldsymbol{\Sigma}_t(\boldsymbol{x}_t) = \boldsymbol{U}_t \,\mathrm{diag}(\lambda_1^t, \cdots, \lambda_n^t)\boldsymbol{U}_t^{\mathsf{T}}$, with nonnegative eigenvalues $\lambda_k^t$. Since $\mathrm{tr}(\boldsymbol{\Sigma}_t) = \sum_k \lambda_k^t$, the MSE corresponds to the sum of eigenvalues—the total uncertainty across all principal directions at noise level $t$:

$$\mathrm{MSE}(\mathsf{D}_p, t) = \mathbb{E}_{\boldsymbol{x}_t} \left[ \mathrm{tr}\left( \mathrm{Cov}_p[\boldsymbol{x}|\boldsymbol{x}_t] \right) \right] = \mathbb{E}_{\boldsymbol{x}_t} \left[ \mathrm{tr}\left( \boldsymbol{\Sigma}_t(\boldsymbol{x}_t) \right) \right] = \mathbb{E}_{\boldsymbol{x}_t} \left[ \sum_{k=1}^n \lambda_k^t(\boldsymbol{x}_t) \right]. \tag{8}$$

For InD samples, the spectrum is compact with smaller leading eigenvalues, reflecting structured denoising aligned with the training data. For OOD samples, uncertainty spreads across multiple eigen-directions, inflating both the spectrum and the trace. This spectral inflation provides a quantitative signal of distribution shift. Figure 2 illustrates this effect: OOD samples exhibit consistently higher uncertainty, confirming the link between spectral inflation and distribution mismatch.

## 3.4 EIGENSCORE AS AN OOD METRIC

While Prop.1 links distribution shift to excess denoising error, it requires two denoisers, whereas in practice we have only a single InD model. Combining Prop.1 with Eq. 7 implies that OOD inputs exhibit inflated posterior covariance, which we can read from its eigenvalues. For each input $\boldsymbol{x}$ and diffusion step $t$, we compute

$$m_t(\boldsymbol{x}) = \sum_{k=1}^K \lambda_k^t(\boldsymbol{x}_t), \quad \boldsymbol{x}_t = \boldsymbol{x} + \sigma_t \boldsymbol{z}, \ \ \boldsymbol{z} \sim \mathcal{N}(0, \boldsymbol{I}), \tag{9}$$

the sum of the top–$K$ eigenvalues across $I$ noise realizations. These are aggregated (e.g., by mean, median, or full set) into $\overline{m}_t(\boldsymbol{x})$, yielding the EigenScore feature vector

$$\mathbf{M}(\boldsymbol{x}) = \left[ \overline{m}_1(\boldsymbol{x}), \ldots, \overline{m}_T(\boldsymbol{x}) \right]^{\mathsf{T}}. \tag{10}$$

We normalize each coordinate using Z-scores with statistics $(\mu_t, \sigma_t)$ computed on training set $z_t(\boldsymbol{x}) = (\overline{m}_t(\boldsymbol{x}) - \mu_t)/\sigma_t$. The final OOD score is the sum of normalized coordinates, with the number of timesteps $T$ tuned for efficiency on validation data.

Direct Jacobian evaluation is costly. Following (Manor & Michaeli, 2024), we estimate the top–$K$ eigenvalues using a Jacobian-free subspace iteration. The method approximates Jacobian–vector products with finite differences of the denoiser $\boldsymbol{v}^+ \approx (\mathsf{D}(\boldsymbol{x}_t + c\boldsymbol{v}) - \mathsf{D}(\boldsymbol{x}_t - c\boldsymbol{v}))/2c$, where $c \ll 1$ is the linear approximation constant, $\boldsymbol{v}$ is the current principle component, and $\boldsymbol{v}^+$ in the next principle component. We then orthogonalizes directions via QR. After a few iterations, eigenvalues are obtained as

$$\lambda_k^t(\boldsymbol{x}_t) \approx \frac{\sigma_t^2}{2c} \left\| \mathsf{D}(\boldsymbol{x}_t + c\boldsymbol{v}^k) - \mathsf{D}(\boldsymbol{x}_t - c\boldsymbol{v}^k) \right\|_2. \tag{11}$$

Full pseudo-code is given in App. B.1.

### 3.5 EIGENSCORE VS. MSE: CAPTURING STRUCTURE INSTEAD OF COLLAPSE

Eq. (4) and Prop. 1 show that distribution shift manifests as excess denoising error, measurable through the posterior covariance. However, scalar metrics such as MSE or score norms collapse this uncertainty into a single number, discarding how variance is distributed across directions. At high noise levels, this collapse is particularly problematic: many small eigenvalues are dominated by isotropic noise, obscuring class-specific structure. The following lemma formalizes this effect.

**Lemma 1.** *Let $p(\boldsymbol{x}_t) = p * \mathcal{N}(0, \sigma_t^2 \boldsymbol{I})$ denotes the noisy marginal in Eq. (3) and let $\boldsymbol{\Sigma}_t(\boldsymbol{x}_t) = \sigma_t^2(\boldsymbol{I} + \sigma_t^2 \nabla^2 \log p(\boldsymbol{x}_t))$ from Eq. (6). As $\sigma_t \to \infty$, $\|\nabla^2 \log p(\boldsymbol{x}_t)\| \to 0$ uniformly on compact sets. Hence*

$$\boldsymbol{\Sigma}_t(\boldsymbol{x}_t) = \sigma_t^2 \boldsymbol{I} + o(\sigma_t^2),$$

*so all eigenvalues satisfy $\lambda_k^t(\boldsymbol{x}_t) = \sigma_t^2 + o(\sigma_t^2)$.*

Lemma 1 implies that the spectrum flattens under heavy Gaussian smoothing: all directions approach the same variance $\sigma_t^2$, so low-variance components lose discriminative information. Consequently, MSE and score norms aggregate mostly isotropic noise, rather than meaningful structure (proof in App. A.4). To retain the informative structure, we focus on the dominant modes. Ky Fan's theorem guarantees that the top-$K$ eigenvalues capture the maximal variance among all $K$-dimensional projections:

**Proposition 2** (**Ky Fan's theorem** (Fan, 1950)). *Let $\boldsymbol{\Sigma}_t(\boldsymbol{x}_t) \succeq 0$ have eigenvalues $\lambda_1^t \geq \cdots \geq \lambda_n^t$ with eigenvectors forming $\boldsymbol{U}_t = [\boldsymbol{u}_1^t, \ldots, \boldsymbol{u}_K^t]$. For any $K \in \{1, \ldots, n\}$,*

$$\max_{\boldsymbol{V} \in \mathbb{R}^{n \times K}: \boldsymbol{V}^\top \boldsymbol{V} = \boldsymbol{I}_K} \operatorname{tr}\left(\boldsymbol{V}^\top \boldsymbol{\Sigma}_t(\boldsymbol{x}_t)\boldsymbol{V}\right) = \sum_{k=1}^{K} \lambda_k,$$

*and a maximizer is $\boldsymbol{V}^\star = [\boldsymbol{u}_1^t, \ldots, \boldsymbol{u}_K^t]$.*

By retaining only the top-$K$ eigenvalues, EigenScore preserves the most informative uncertainty directions while discarding noise-dominated components. This explains its consistent advantage over MSE and score-based metrics. The choice of $K$ is not dictated by theory, but reflects a practical trade-off between capturing discriminative information and computational efficiency (see App. A.5 for proof). Reconstruction-based diffusion methods such as DDPM-OOD (Graham et al., 2023) can be interpreted as relying on scalarized denoising error, either in pixel space or perceptual feature space. As with MSE, these approaches collapse uncertainty into a single value, discarding how variance is distributed across directions. EigenScore differs fundamentally by retaining the dominant spectral modes of posterior covariance, which remain discriminative even when reconstruction error alone becomes unreliable. We further discuss the implications of using learned denoisers, whose Jacobians need not be strictly SPD, and their relation to spectral truncation in Appendix B.2.

## 4 EXPERIMENTS

We now evaluate the effectiveness of EigenScore for OOD detection. Specifically, we benchmark EigenScore across a suite of pairwise OOD detection tasks and compare its performance against state-of-the-art baselines.

**Datasets.** We evaluate OOD detection on standard image benchmarks commonly used with diffusion models: C10 (Krizhevsky, 2009), SVHN (Netzer et al., 2011), C100 (Krizhevsky, 2009), and CelebA (Liu et al., 2015a;b). For near-OOD tasks (Yang et al., 2022), we additionally include TinyImageNet (Le & Yang, 2015). Further details can be found in Appendix (B.5)

Table 1: *Main OOD detection results (AUROC). Comparison of EigenScore with likelihood-based, reconstruction-based, and diffusion-based baselines across multiple InD–OOD dataset pairings (CelebA, C10, C100, SVHN).* **Best** *and* **second best** *are highlighted. Note that EigenScore achieves the best average performance and is either best or second best in most settings.*

| InD | **CelebA** vs. | | | **C10** vs. | | | **C100** vs. | | | **SVHN** vs. | | | |
|-----|------|------|------|------|------|--------|------|--------|------|------|------|--------|------|
| OOD | C10 | C100 | SVHN | C100 | SVHN | CelebA | C10 | CelebA | SVHN | C10 | C100 | CelebA | **Avg** |
| DoS | 0.630 | 0.615 | 0.808 | 0.504 | 0.752 | 0.456 | 0.491 | 0.520 | 0.777 | 0.911 | 0.904 | 0.956 | 0.693 |
| TT | 0.676 | 0.655 | 0.773 | 0.558 | 0.714 | 0.469 | 0.538 | 0.464 | 0.648 | 0.957 | 0.961 | 0.994 | 0.701 |
| WAIC | 0.589 | 0.569 | 0.793 | 0.476 | 0.760 | 0.469 | 0.502 | 0.530 | **0.782** | **0.978** | **0.974** | 0.955 | 0.698 |
| *Diffusion-based* | | | | | | | | | | | | | |
| NLL | 0.507 | 0.671 | 0.753 | 0.558 | 0.545 | 0.599 | 0.480 | 0.484 | 0.481 | 0.635 | 0.660 | 0.636 | 0.584 |
| IC | 0.510 | 0.673 | 0.755 | 0.552 | 0.540 | 0.583 | 0.460 | 0.466 | 0.469 | 0.625 | 0.653 | 0.625 | 0.576 |
| DDPM-OOD | 0.922 | 0.928 | **0.992** | **0.618** | **0.944** | 0.642 | 0.462 | 0.496 | **0.870** | 0.963 | 0.972 | **0.996** | **0.817** |
| LMD | 0.886 | 0.848 | 0.950 | 0.601 | **0.821** | 0.834 | **0.569** | **0.595** | 0.748 | 0.780 | 0.749 | 0.872 | 0.771 |
| DiffPath (CelebA) | **1.000** | **1.000** | 0.964 | 0.554 | 0.729 | **0.885** | 0.475 | **0.887** | 0.724 | 0.919 | 0.941 | 0.328 | 0.784 |
| DiffPathV2 (CelebA) | **1.000** | 0.995 | **0.969** | 0.535 | 0.812 | 0.862 | 0.483 | 0.513 | 0.724 | **0.969** | **0.975** | 0.883 | 0.810 |
| **EigenScore (Ours)** | **0.965** | **0.944** | 0.888 | **0.880** | 0.810 | 0.873 | **0.642** | 0.427 | 0.661 | **0.992** | **0.982** | **0.994** | **0.838** |

**Baselines.** We compare our method against several generative baselines for OOD detection, including Improved CD (Du et al., 2021), DoS (Morningstar et al., 2021), TT (Nalisnick et al., 2019b), WAIC (Choi et al., 2018), NLL, IC, DDPM-OOD (Graham et al., 2023), LMD (Liu et al., 2023), DiffPathV2 (Abdi et al., 2025), and DiffPath (Heng et al., 2024). Details regarding the baselines can be found in Appendix (B.4).

Table 2: *Near-OOD detection results (AUROC). We evaluate on semantically related datasets, including C10 vs. C100 and TinyImageNet, which are particularly challenging due to shared low-level statistics between InD and OOD samples. The* **best** *and* **second best** *methods are highlighted. EigenScore achieves the best average performance across both tasks, with a clear margin over prior diffusion-based approaches.*

| InD | **C10** vs. | | **C100** vs. | | |
|-----|------|--------------|------|--------------|------|
| OOD | C100 | TinyImageNet | C10 | TinyImageNet | **Avg** |
| DDPM-OOD | **0.618** | 0.570 | 0.462 | 0.457 | 0.527 |
| LMD | 0.601 | 0.592 | **0.569** | 0.558 | 0.580 |
| DiffPath | 0.554 | **0.993** | 0.475 | **0.995** | **0.754** |
| **EigenScore** | **0.884** | **0.973** | **0.652** | **0.888** | **0.849** |

## 4.1 MAIN RESULTS

Table 1 reports AUROC results across all dataset pairs. EigenScore achieves the highest *average* performance across all settings and is best or second best on nearly every dataset pair. Its advantage is most pronounced in the challenging near–OOD regime (CIFAR-10 vs. CIFAR-100), where likelihood-based scores and trajectory metrics often fail. For example, EigenScore improves AUROC by up to 2% over the best diffusion-based baseline, consistent with our theoretical claim that retaining leading eigenvalues preserves discriminative structure.

On the near-OOD task (Yang et al., 2022) (C10 vs. C100/TinyImageNet), EigenScore continues to deliver strong separation, achieving higher average AUROC than other diffusion-based metrics in Table 2. In particular, while baselines such as NLL, IC, and WAIC (C10 vs. C100) struggle to distinguish the closely related distributions, EigenScore consistently maintains reliable performance, showing its robustness even under challenging near-OOD settings. Notably, EigenScore outperforms DDPM-OOD, where reconstruction quality remains high but posterior uncertainty inflates.

## 4.2 ABLATIONS

**Number of timesteps.** The parameter $T$ determines how many points along the diffusion trajectory contribute to the score. Larger $T$ includes more noise levels but increases computation and eventually saturates, since high noise merely lifts all eigenvalues uniformly (Lemma 1) without adding discriminative power. As shown in Table 3, even a small budget (e.g., $T=5$) achieves nearly the same AUROC as larger $T$, with only marginal improvements from denser schedules. This confirms that a compact subset of timesteps captures most of the useful information, balancing accuracy and efficiency.

Table 3: *Ablation on timesteps $T$, repetitions $I$, and number of eigenvalues $K$. AUROC performance of EigenScore across InD–OOD dataset pairs.*

| Timesteps | CelebA vs. | | | C10 vs. | | | C100 vs. | | | SVHN vs. | | | Avg |
|---|---|---|---|---|---|---|---|---|---|---|---|---|---|
| | C10 | C100 | SVHN | C100 | SVHN | CelebA | C10 | CelebA | SVHN | C10 | C100 | CelebA | |
| 5 | **0.964** | **0.943** | **0.893** | **0.869** | **0.778** | 0.866 | **0.635** | 0.448 | **0.648** | **0.990** | **0.977** | 0.995 | **0.834** |
| 7 | 0.963 | 0.942 | 0.878 | 0.848 | 0.703 | 0.862 | 0.619 | 0.500 | 0.603 | 0.988 | 0.975 | 0.996 | 0.823 |
| 10 | 0.952 | 0.931 | 0.850 | 0.819 | 0.627 | **0.867** | 0.578 | **0.589** | 0.521 | 0.987 | 0.975 | **0.998** | 0.808 |

| Repetitions $I$ | CelebA vs. | | | C10 vs. | | | C100 vs. | | | SVHN vs. | | | Avg |
|---|---|---|---|---|---|---|---|---|---|---|---|---|---|
| | C10 | C100 | SVHN | C100 | SVHN | CelebA | C10 | CelebA | SVHN | C10 | C100 | CelebA | |
| 5 | 0.959 | 0.940 | 0.885 | 0.863 | 0.773 | 0.857 | **0.635** | **0.455** | 0.649 | **0.990** | 0.977 | **0.995** | 0.832 |
| 10 | 0.962 | 0.942 | 0.889 | 0.867 | 0.776 | 0.864 | **0.635** | 0.450 | **0.650** | **0.990** | 0.978 | **0.995** | 0.833 |
| 15 | 0.962 | 0.942 | 0.891 | **0.869** | **0.778** | 0.866 | **0.635** | 0.447 | 0.649 | **0.990** | 0.977 | 0.994 | 0.833 |
| 20 | **0.964** | **0.943** | **0.893** | **0.869** | **0.778** | **0.866** | **0.635** | 0.448 | 0.648 | **0.990** | 0.977 | **0.995** | **0.834** |

| Eigenvalues $K$ | CelebA vs. | | | C10 vs. | | | C100 vs. | | | SVHN vs. | | | Avg |
|---|---|---|---|---|---|---|---|---|---|---|---|---|---|
| | C10 | C100 | SVHN | C100 | SVHN | CelebA | C10 | CelebA | SVHN | C10 | C100 | CelebA | |
| 1 | **0.968** | **0.950** | **0.945** | 0.871 | **0.803** | 0.830 | **0.639** | 0.432 | **0.713** | 0.983 | 0.966 | 0.983 | **0.840** |
| 2 | 0.967 | 0.946 | 0.919 | **0.872** | 0.793 | 0.852 | 0.636 | 0.439 | 0.679 | 0.987 | 0.972 | 0.991 | 0.838 |
| 3 | 0.964 | 0.943 | 0.893 | 0.869 | 0.778 | **0.866** | 0.635 | **0.448** | 0.648 | **0.990** | **0.977** | **0.995** | 0.834 |

Table 4: *Comparison of MSE vs. EigenScore. Average AUROC across all InD–OOD dataset pairs. EigenScore consistently outperforms MSE by leveraging spectral structure of the uncertainty.*

| Method | CelebA vs. | | | C10 vs. | | | C100 vs. | | | SVHN vs. | | | Avg |
|---|---|---|---|---|---|---|---|---|---|---|---|---|---|
| | C10 | C100 | SVHN | C100 | SVHN | CelebA | C10 | CelebA | SVHN | C10 | C100 | CelebA | |
| MSE | 0.804 | 0.783 | 0.220 | 0.629 | 0.184 | 0.841 | 0.552 | **0.675** | 0.147 | **0.994** | **0.994** | **1.000** | 0.652 |
| EigenScore | **0.964** | **0.943** | **0.893** | **0.869** | **0.778** | **0.866** | **0.635** | 0.448 | **0.648** | 0.990 | 0.977 | 0.995 | **0.834** |

**Number of repetitions.** The parameter $I$ sets how many noise draws are averaged per timestep. Larger $I$ reduces variance in the estimated eigenvalues and stabilizes OOD scores, but also increases runtime. As shown in Table 3, small values (e.g., $I{=}5$) are already sufficient, with only marginal gains beyond $I{=}15$. The results are reported with timesteps $t \in \{100, 150, 200, 250, 300\}$, mean aggregation, and $K{=}3$ eigenvalues.

**Number of eigenvalues.** The parameter $K$ determines how many leading eigenvalues are aggregated at each timestep. Table 3 shows that $K{=}1$ achieves the best average performance, though in some settings $K{=}3$ yields slightly higher AUROC. This indicates that the bulk of discriminative information resides in the first few modes, consistent with Lemma 1. The results are reported with the same timestep schedule, mean aggregation, and $I{=}20$ repetitions.

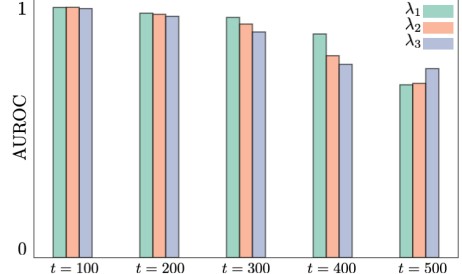

Figure 3: *Ablation on eigenvalue informativeness across $t$. Performance declines with increasing noise, consistent with Lem. 1, while $\lambda_1$ retains the strongest OOD signal compared to $\lambda_2$ and $\lambda_3$, supporting Prop. 2.*

**EigenScore vs. MSE.** EigenScore consistently outperforms MSE by retaining the spectral structure of posterior covariance rather than collapsing uncertainty into a single scalar. As shown in Table 4, this leads to higher AUROC across diverse dataset pairs. All experiments here use 20 repetitions, timesteps $t \in \{100, 150, 200, 250, 300\}$, and mean aggregation.

To further examine Lemma 1 and Prop. 2, we analyze the informativeness of individual eigenvalues across noise levels. Lemma 1 predicts that at larger $t$, eigenvalues converge toward $\sigma_t^2$, diminishing their discriminative value. Figure 3 confirms this: AUROC with $\lambda_1$ decreases gradually as $t$ increases, while $\lambda_2$ and $\lambda_3$ deteriorate more sharply. Prop. 2 further implies that the leading eigenvalues capture the most informative variance. Consistently, $\lambda_1$ provides the strongest OOD signal, followed by $\lambda_2$ and then $\lambda_3$. These results highlight why focusing on dominant modes, as done in EigenScore, yields more stable and informative detection than scalarized measures such as MSE.

## 5 CONCLUSION

We introduced EigenScore, a principled OOD detection method for diffusion models that leverages the spectral structure of denoising uncertainty. By linking KL divergence to excess denoising error and showing that posterior covariance inflation consistently signals distribution shift, EigenScore offers both theoretical justification and strong empirical performance. Across diverse benchmarks, EigenScore consistently outperformed likelihood-based and score-norm methods, with particularly robust gains in near-OOD settings where traditional approaches fail. Ablation studies further confirmed that most discriminative information lies in the leading eigenvalues at moderate noise levels, validating our spectral perspective.

## LIMITATIONS

Despite its strengths, our approach has several limitations. First, EigenScore only leverages a subset of the information available in diffusion models: we use the eigenvalues of the posterior covariance but discard eigenvector structure, which may contain additional discriminative cues. Moreover, we compute features at a limited set of timesteps rather than across the full diffusion trajectory, potentially overlooking temporal dynamics of uncertainty. Second, our framework focuses on the magnitude of eigenvalues, but does not explicitly exploit their rate of change across eigenvalues, which itself may differ systematically between InD and OOD inputs and could serve as a detection signal.

## REPRODUCIBILITY STATEMENT

We have taken several steps to ensure the reproducibility of our results. All datasets used in our experiments (CIFAR-10, CIFAR-100, SVHN, CelebA, TinyImageNet) are publicly available. We provide detailed descriptions of our training and evaluation protocols, including the diffusion model architecture, noise schedules, hyperparameters, and aggregation strategies. Experimental results are averaged across multiple random seeds, and we report the effect of varying key parameters (number of timesteps, repetitions, and eigenvalues) in Section 4.2. To further facilitate reproducibility, we have released the full anonymous source code and scripts for running experiments in supplement B.5.

## LLMS USAGE STATEMENT

During the preparation of this manuscript, we made limited use of large language models (LLMs), specifically OpenAI's ChatGPT, to assist with language refinement and organization of some sections. All technical content, equations, derivations, and experimental design were developed entirely by the authors. The LLM was not used for ideation of methods, data analysis, or generation of results.

## ACKNOWLEDGMENTS

This paper is partially based on work supported by the NSF under CAREER Awards CCF-2043134, 2046293 and Grants CCF-2504613, 2504614.

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

# A    PROOFS

## A.1    CONNECTING KL DIVERGENCE WITH DENOISING ERROR

Eq. (4) is inspired from the work of Song et al. (2021) [Theorem 1] and Shoushtari et al. (2025) [Theorem 1], where KL divergence between two distribution is stated in terms of fisher divergence:

$$D_{\mathrm{KL}}(p \parallel q) = \int_0^\infty \mathbb{E}_{\boldsymbol{x},\boldsymbol{x}_t} \left[ \left\| \nabla \log p(\boldsymbol{x}_t) - \nabla \log q(\boldsymbol{x}_t) \right\|_2^2 \right] \sigma_t \, \mathrm{d}t. \tag{12}$$

By using the Tweedie's formula from Eq. (2)

$$D_p(\boldsymbol{x}_t) = \mathbb{E}_p[\boldsymbol{x}|\boldsymbol{x}_t] = \boldsymbol{x}_t + \sigma_t^2 \nabla \log p(\boldsymbol{x}_t),$$

and replacing the score functions for distributions $p$ and $q$ with their corresponding MMSE estimators, we have:

$$D_{\mathrm{KL}}(p \parallel q) = \int_0^\infty \mathbb{E}_{\boldsymbol{x},\boldsymbol{x}_t} \left[ \left\| \mathbb{E}_p[\boldsymbol{x}|\boldsymbol{x}_t] - \mathbb{E}_q[\boldsymbol{x}|\boldsymbol{x}_t] \right\|_2^2 \right] \sigma_t^{-3} \, \mathrm{d}t$$

$$= \int_0^\infty \left\{ \mathbb{E}_{\boldsymbol{x},\boldsymbol{x}_t} \left[ \left\| \boldsymbol{x} - \mathbb{E}_q[\boldsymbol{x}|\boldsymbol{x}_t] \right\|_2^2 \right] - \mathbb{E}_{\boldsymbol{x},\boldsymbol{x}_t} \left[ \left\| \boldsymbol{x} - \mathbb{E}_p[\boldsymbol{x}|\boldsymbol{x}_t] \right\|_2^2 \right] \right\} \sigma_t^{-3} \, \mathrm{d}t, \tag{13}$$

where in the second line, we used the following decomposition:

$$\mathbb{E}_{\boldsymbol{x},\boldsymbol{x}_t} \left[ \left\| \boldsymbol{x} - \mathbb{E}_q[\boldsymbol{x}|\boldsymbol{x}_t] \right\|_2^2 \right] = \mathbb{E}_{\boldsymbol{x},\boldsymbol{x}_t} \left[ \left\| \boldsymbol{x} - \mathbb{E}_p[\boldsymbol{x}|\boldsymbol{x}_t] + \mathbb{E}_p[\boldsymbol{x}|\boldsymbol{x}_t] - \mathbb{E}_q[\boldsymbol{x}|\boldsymbol{x}_t] \right\|_2^2 \right]$$

$$= \mathbb{E}_{\boldsymbol{x},\boldsymbol{x}_t} \left[ \left\| \boldsymbol{x} - \mathbb{E}_p[\boldsymbol{x}|\boldsymbol{x}_t] \right\|_2^2 \right] + \mathbb{E}_{\boldsymbol{x},\boldsymbol{x}_t} \left[ \left\| \mathbb{E}_p[\boldsymbol{x}|\boldsymbol{x}_t] - \mathbb{E}_q[\boldsymbol{x}|\boldsymbol{x}_t] \right\|_2^2 \right]$$

$$+ 2\mathbb{E}_{\boldsymbol{x},\boldsymbol{x}_t} \left[ \left( \boldsymbol{x} - \mathbb{E}_p[\boldsymbol{x}|\boldsymbol{x}_t] \right)^\mathsf{T} \left( \mathbb{E}_p[\boldsymbol{x}|\boldsymbol{x}_t] - \mathbb{E}_q[\boldsymbol{x}|\boldsymbol{x}_t] \right) \right]$$

$$= \mathbb{E}_{\boldsymbol{x},\boldsymbol{x}_t} \left[ \left\| \boldsymbol{x} - \mathbb{E}_p[\boldsymbol{x}|\boldsymbol{x}_t] \right\|_2^2 \right] + \mathbb{E}_{\boldsymbol{x},\boldsymbol{x}_t} \left[ \left\| \mathbb{E}_p[\boldsymbol{x}|\boldsymbol{x}_t] - \mathbb{E}_q[\boldsymbol{x}|\boldsymbol{x}_t] \right\|_2^2 \right],$$

where in the last equality, we used the fact that

$$\mathbb{E}_{\boldsymbol{x},\boldsymbol{x}_t} \left[ \left( \boldsymbol{x} - \mathbb{E}_p[\boldsymbol{x}|\boldsymbol{x}_t] \right) \right] = 0, \quad \text{where} \quad \boldsymbol{x} \sim p(\boldsymbol{x}) \ \text{ and } \ \boldsymbol{x}_t \sim p(\boldsymbol{x}_t).$$

By replacing the result of this equation into Eq. (13)

$$\mathbb{E}_{\boldsymbol{x},\boldsymbol{x}_t} \left[ \left\| \mathbb{E}_p[\boldsymbol{x}|\boldsymbol{x}_t] - \mathbb{E}_q[\boldsymbol{x}|\boldsymbol{x}_t] \right\|_2^2 \right] = \mathbb{E}_{\boldsymbol{x},\boldsymbol{x}_t} \left[ \left\| \boldsymbol{x} - \mathbb{E}_q[\boldsymbol{x}|\boldsymbol{x}_t] \right\|_2^2 \right] - \mathbb{E}_{\boldsymbol{x},\boldsymbol{x}_t} \left[ \left\| \boldsymbol{x} - \mathbb{E}_p[\boldsymbol{x}|\boldsymbol{x}_t] \right\|_2^2 \right]$$

$$= \mathrm{MSE}(D_q, t) - \mathrm{MSE}(D_p, t).$$

## A.2    CONNECTING MSE OF DENOISING TO COVARIANCE

We assume that MMSE estimator $D_p$ is an optimal denoiser trained on images sampled from data distribution $p(\boldsymbol{x})$. Consequently, the MSE for this operator is defined as

$$\mathrm{MSE}(D_p, t) = \mathbb{E}_{\boldsymbol{x},\boldsymbol{x}_t} \left[ \left\| \boldsymbol{x} - \mathbb{E}_p[\boldsymbol{x}|\boldsymbol{x}_t] \right\|_2^2 \right] = \mathbb{E}_{\boldsymbol{x}} \left[ \mathbb{E}_{\boldsymbol{x}_t} \left[ \mathrm{tr}(\boldsymbol{x} - \mathbb{E}_p[\boldsymbol{x}|\boldsymbol{x}_t])(\boldsymbol{x} - \mathbb{E}_p[\boldsymbol{x}|\boldsymbol{x}_t])^\mathsf{T} \big| \boldsymbol{x}_t \right] \right]$$

$$= \mathbb{E}_{\boldsymbol{x}} \left[ \mathrm{tr}\, \mathrm{Cov}_p[\boldsymbol{x}|\boldsymbol{x}_t] \right],$$

where the first equality follows from the definition of MSE using the MMSE estimator $D_p(\boldsymbol{x}_t) = \mathbb{E}_p[\boldsymbol{x}|\boldsymbol{x}_t]$, and the second equality applies the law of total expectation along with the identity $\|\boldsymbol{v}\|_2^2 = \mathrm{tr}(\boldsymbol{v}\boldsymbol{v}^\mathsf{T})$. In the last equality, we used the fact that the inner expectation corresponds to the conditional covariance matrix $\mathrm{Cov}[\boldsymbol{x}|\boldsymbol{x}_t]$, whose trace gives the expected squared error. From Theorem 1 of Manor & Michaeli (2024), we have

$$\mathrm{Cov}_p[\boldsymbol{x}|\boldsymbol{x}_t] = \sigma_t^2 \nabla D_p(\boldsymbol{x}_t) \approx \sum_{k=1}^K \lambda_k^t(\boldsymbol{x}_t) \boldsymbol{u}_k \boldsymbol{u}_k^\mathsf{T}, \tag{14}$$

where $\nabla D_p$ is the Jacobian, assumed to be positive semi-definite and symmetric. We approximate the Jacobian matrix using the top $K$ eigenvalues $\lambda_1, \cdots, \lambda_K$, nd their corresponding eigenvectors $u_1, \cdots, u_K$ which is a low-rank approximation of the full covariance matrix based on its spectral decomposition of symmetric positive semi-definite matrices. By retaining only the top $K < n$ terms in this expansion, we obtain a rank-$K$ approximation of the covariance, capturing most of its variance while reducing dimensionality and computation. The trace of this approximate covariance is then given by

$$\operatorname{tr}(\operatorname{Cov}_p[\boldsymbol{x}|\boldsymbol{x}_t]) \approx \sum_{k=1}^{K} \lambda_k^t(\boldsymbol{x}_t), \tag{15}$$

since each term $u_k u_k^{\mathsf{T}}$ is a rank-one projection matrix with unit trace. This expression allows us to estimate the total variance (or the MSE) using only the top eigenvalues, under the assumption that the remaining eigenvalues contribute negligibly. Consequently, we have:

$$\operatorname{MSE}(D_p, t) = \mathbb{E}_{\boldsymbol{x}}\left[\operatorname{tr}\operatorname{Cov}_p[\boldsymbol{x}|\boldsymbol{x}_t]\right] = \sigma_t^2 \sum_{k=1}^{K} \lambda_k^t(\boldsymbol{x}_t). \tag{16}$$

### A.3 MIYASAWA RELATIONSHIP

The connection between MMSE estimation under Gaussian noise and the score function was first established in Miyasawa (1961) and later generalized in Raphan & Simoncelli (2011). For completeness, we provide a derivation here.

Let $\boldsymbol{x}_t = \boldsymbol{x} + \boldsymbol{z}$, denote a noisy observation of $\boldsymbol{x} \sim p(\boldsymbol{x})$, where $\boldsymbol{z} \sim \mathcal{N}(0, \sigma_t^2 \boldsymbol{I})$, and define $p(\boldsymbol{x}_t|\boldsymbol{x})$ as the Gaussian likelihood. The marginal distribution of y is given by:

$$p(\boldsymbol{x}_t) = \int p(\boldsymbol{x})\, p(\boldsymbol{x}_t|\boldsymbol{x})\, \mathrm{d}\boldsymbol{x}.$$

To derive the score function $\nabla_{\boldsymbol{x}_t} \log p(\boldsymbol{x}_t)$ , we differentiate the log-marginal using the identity $\nabla h(\boldsymbol{x}_t) = h(\boldsymbol{x}_t)\nabla \log h(\boldsymbol{x}_t)$:

$$\nabla_{\boldsymbol{x}_t} \log p(\boldsymbol{x}_t) = \int p(\boldsymbol{x})p(\boldsymbol{x}_t|\boldsymbol{x})\nabla_{\boldsymbol{x}_t} \log p(\boldsymbol{x}_t|\boldsymbol{x})\mathrm{d}\boldsymbol{x} \Big/ p(\boldsymbol{x}_t)$$

$$= \int p(\boldsymbol{x}|\boldsymbol{x}_t)\nabla_{\boldsymbol{x}_t} \log p(\boldsymbol{x}_t|\boldsymbol{x})\mathrm{d}\boldsymbol{x}$$

$$= \mathbb{E}\left[\nabla_{\boldsymbol{x}_t} \log p(\boldsymbol{x}_t|\boldsymbol{x}) \mid \boldsymbol{x}_t\right],$$

which can be interpreted as a chain rule applied to score functions rather than densities.

Next, we compute the Hessian $\nabla^2 \log p(\boldsymbol{x}_t)$. Differentiating again:

$$\nabla^2 \log p(\boldsymbol{x}_t) = \int p(\boldsymbol{x}|\boldsymbol{x}_t)\left(\nabla_{\boldsymbol{x}_t} \log p(\boldsymbol{x}|\boldsymbol{x}_t)\nabla_{\boldsymbol{x}_t} \log p(\boldsymbol{x}_t|\boldsymbol{x})^{\mathsf{T}} + \nabla_{\boldsymbol{x}_t}^2 \log p(\boldsymbol{x}_t|\boldsymbol{x})\right)\mathrm{d}\boldsymbol{x}.$$

Using Bayes' rule:

$$\log p(\boldsymbol{x}|\boldsymbol{x}_t) = \log p(\boldsymbol{x}_t|\boldsymbol{x}) + \log p(\boldsymbol{x}) - \log p(\boldsymbol{x}_t),$$

and differentiating with respect to y, we obtain:

$$\nabla_{\boldsymbol{x}_t} \log p(\boldsymbol{x}|\boldsymbol{x}_t) = \nabla_{\boldsymbol{x}_t} \log p(\boldsymbol{x}_t|\boldsymbol{x}) - \nabla \log p(\boldsymbol{x}_t).$$

Substituting this into the previous expression yields:

$$\nabla^2 \log p(\boldsymbol{x}_t) = \int p(\boldsymbol{x}|\boldsymbol{x}_t)\left((\nabla_{\boldsymbol{x}_t} \log p(\boldsymbol{x}_t|\boldsymbol{x}) - \nabla \log p(\boldsymbol{x}_t))\nabla_{\boldsymbol{x}_t} \log p(\boldsymbol{x}_t|\boldsymbol{x})^{\mathsf{T}} + \nabla_{\boldsymbol{x}_t}^2 \log p(\boldsymbol{x}_t|\boldsymbol{x})\right)\mathrm{d}\boldsymbol{x}$$

$$= \mathbb{E}\left[(\nabla_{\boldsymbol{x}_t} \log p(\boldsymbol{x}_t|\boldsymbol{x}) - \nabla \log p(\boldsymbol{x}_t))\nabla_{\boldsymbol{x}_t} \log p(\boldsymbol{x}_t|\boldsymbol{x})^{\mathsf{T}} \mid \boldsymbol{x}_t\right] + \mathbb{E}\left[\nabla_{\boldsymbol{x}_t}^2 \log p(\boldsymbol{x}_t|\boldsymbol{x}) \mid \boldsymbol{x}_t\right]$$

$$= \operatorname{Cov}\left[\nabla_{\boldsymbol{x}_t} \log p(\boldsymbol{x}_t|\boldsymbol{x}) \mid \boldsymbol{x}_t\right] + \mathbb{E}\left[\nabla_{\boldsymbol{x}_t}^2 \log p(\boldsymbol{x}_t|\boldsymbol{x}) \mid \boldsymbol{x}_t\right].$$

Now, since $p(\boldsymbol{x}_t|\boldsymbol{x})$ is Gaussian with variance $\sigma_t^2 \boldsymbol{I}$, we have:

$$\log p(\boldsymbol{x}_t|\boldsymbol{x}) = -\frac{1}{2\sigma_t^2}\|\boldsymbol{x}_t - \boldsymbol{x}\|^2 + \text{const},$$

$$\nabla_{\boldsymbol{x}_t} \log p(\boldsymbol{x}_t|\boldsymbol{x}) = -\frac{1}{\sigma_t^2}(\boldsymbol{x}_t - \boldsymbol{x}),$$

$$\nabla_{\boldsymbol{x}_t} \log p(\boldsymbol{x}_t|\boldsymbol{x}) = -\frac{1}{\sigma_t^2}\boldsymbol{I}.$$

Plugging into the previous results gives the Miyasawa identities:

$$\nabla \log p(\boldsymbol{x}_t) = \frac{1}{\sigma_t^2}\left(\mathbb{E}[\boldsymbol{x} \mid \boldsymbol{x}_t] - \boldsymbol{x}_t\right), \tag{17}$$

$$\nabla^2 \log p(\boldsymbol{x}_t) = \frac{1}{\sigma_t^4}\text{Cov}[\boldsymbol{x} \mid \boldsymbol{x}_t] - \frac{1}{\sigma_t^2}\boldsymbol{I}. \tag{18}$$

Rearranging, we recover both the posterior mean and posterior covariance in terms of the score and its Hessian:

$$\mathbb{E}[\boldsymbol{x} \mid \boldsymbol{x}_t] = \boldsymbol{x}_t + \sigma_t^2 \nabla \log p(\boldsymbol{x}_t), \tag{19}$$

$$\text{Cov}[\boldsymbol{x} \mid \boldsymbol{x}_t] = \sigma_t^2\left(\boldsymbol{I} + \sigma_t^2 \nabla^2 \log p(\boldsymbol{x}_t)\right). \tag{20}$$

Finally, the optimal denoising error is given by the expected trace of the posterior covariance:

$$\mathbb{E}\left[\|\boldsymbol{x} - \mathbb{E}[\boldsymbol{x} \mid \boldsymbol{x}_t]\|^2\right] = \mathbb{E}\left[\text{tr}\,\text{Cov}[\boldsymbol{x} \mid \boldsymbol{x}_t]\right].$$

## A.4 PROOF OF LEMMA 1.

**Lemma 1** *Let $p(\boldsymbol{x}_t) = p * \mathcal{N}(0, \sigma_t^2 \boldsymbol{I})$ denotes the noisy marginal distribution in Eq. (3) and $\boldsymbol{\Sigma}_t(\boldsymbol{x}_t) = \sigma_t^2\left(\boldsymbol{I} + \sigma_t^2 \nabla^2 \log p(\boldsymbol{x}_t)\right)$ from Miyasawa (Eq. (6)). As $\sigma_t \to \infty$, $\|\nabla^2 \log p(\boldsymbol{x}_t)\| \to 0$ uniformly on compact sets. Hence*

$$\boldsymbol{\Sigma}_t(\boldsymbol{x}_t) = \sigma_t^2 \boldsymbol{I} + o(\sigma_t^2),$$

*so all eigenvalues satisfy $\lambda_k^t(\boldsymbol{x}_t) = \sigma_t^2 + o(\sigma_t^2)$.*

*Proof.* Let $G_{\sigma_t}$ denotes the Gaussian density function with standard deviation $\sigma_t \geq 0$. Then, we have

$$p(\boldsymbol{x}_t) = \int p(\boldsymbol{x}_t|\boldsymbol{x})p(\boldsymbol{x})\mathrm{d}\boldsymbol{x} = \int G_{\sigma_t}(\boldsymbol{x}_t - \boldsymbol{x})p(\boldsymbol{x})\mathrm{d}\boldsymbol{x}, \tag{21}$$

Differentiation under the integral yields

$$\nabla G_\sigma(\boldsymbol{z}) = -\frac{\boldsymbol{z}}{\sigma^2}G_\sigma(\boldsymbol{z}), \qquad \nabla^2 G_\sigma(\boldsymbol{z}) = \left(\frac{\boldsymbol{z}\boldsymbol{z}^\mathsf{T}}{\sigma^4} - \frac{\boldsymbol{I}}{\sigma^2}\right)G_\sigma(\boldsymbol{z}). \tag{22}$$

Hence

$$\nabla_{\boldsymbol{x}_t} p(\boldsymbol{x}_t) = \int \nabla G_{\sigma_t}(\boldsymbol{x}_t - \boldsymbol{x})\,p(\boldsymbol{x})\,\mathrm{d}\boldsymbol{x}, \qquad \nabla_{\boldsymbol{x}_t}^2 p(\boldsymbol{x}_t) = \int \nabla^2 G_{\sigma_t}(\boldsymbol{x}_t - \boldsymbol{x})\,p(\boldsymbol{x})\,\mathrm{d}\boldsymbol{x}. \tag{23}$$

Fix a compact set $K \subset \mathbb{R}^d$. Using Cauchy–Schwarz and that $p$ has finite second moment, one obtains bounds of the form

$$\frac{\|\nabla p(\boldsymbol{x}_t)\|}{p(\boldsymbol{x}_t)} \leq \frac{C_1}{\sigma_t}, \qquad \frac{\|\nabla^2 p(\boldsymbol{x}_t)\|}{p(\boldsymbol{x}_t)} \leq \frac{C_2}{\sigma_t^2}, \qquad \forall \boldsymbol{x} \in K, \tag{24}$$

for constants $C_1, C_2$ independent of $\sigma_t$. Consequently,

$$\nabla^2 \log p(\boldsymbol{x}_t) = \frac{\nabla^2 p(\boldsymbol{x}_t)}{p(\boldsymbol{x}_t)} - \frac{\nabla p(\boldsymbol{x}_t)\,\nabla p(\boldsymbol{x}_t)^\mathsf{T}}{p(\boldsymbol{x}_t)^2} \tag{25}$$

satisfies

$$\|\nabla^2 \log p(\boldsymbol{x}_t)\| \leq \frac{C_2}{\sigma_t^2} + \frac{C_1^2}{\sigma_t^2} = O\left(\frac{1}{\sigma^2}\right), \tag{26}$$

uniformly on $K$. Thus $\sigma_t^2 \nabla^2 \log p(\boldsymbol{x}_t) \to 0$ as $\sigma_t \to \infty$.

Plugging into

$$\boldsymbol{\Sigma}_t(\boldsymbol{x}_t) = \sigma_t^2(\boldsymbol{I} + \sigma_t^2 \nabla^2 \log p(\boldsymbol{x}_t)) \tag{27}$$

gives

$$\boldsymbol{\Sigma}_t(\boldsymbol{x}_t) = \sigma_t^2 \boldsymbol{I} + o(\sigma_t^2), \tag{28}$$

and hence $\lambda_k^t(\boldsymbol{x}_t) = \sigma_t^2 + o(\sigma_t^2)$ for all $k$. $\square$

This implies that at large noise, the posterior covariance becomes asymptotically isotropic; the spectrum *flattens* and "small" eigenvalues are lifted to $\approx \sigma_t^2$. Their contribution is therefore indistinguishable from isotropic noise, which explains why including late timesteps or many tail eigenvalues does not improve OOD discrimination.

### A.5 PROOF OF KY FAN'S THEOREM

**Prop. 2.** *Let $\boldsymbol{\Sigma}_t(\boldsymbol{x}_t) \succeq 0$ have eigenvalues $\lambda_1^t \geq \cdots \geq \lambda_n^t$ with eigenvectors forming $\boldsymbol{U}_t = [\boldsymbol{u}_1^t, \ldots, \boldsymbol{u}_n^t]$. For any $K \in \{1, \ldots, n\}$,*

$$\max_{\boldsymbol{V} \in \mathbb{R}^{n \times K}: \boldsymbol{V}^\top \boldsymbol{V} = \boldsymbol{I}_K} \mathrm{tr}\left(\boldsymbol{V}^\top \boldsymbol{\Sigma}_t(\boldsymbol{x}_t)\boldsymbol{V}\right) = \sum_{k=1}^K \lambda_k,$$

*and a maximizer is $\boldsymbol{V}^\star = [\boldsymbol{u}_1^t, \ldots, \boldsymbol{u}_K^t]$.*

*Proof.* Let $\boldsymbol{V} \in \mathbb{R}^{n \times K}$ with $\boldsymbol{V}^\top \boldsymbol{V} = \boldsymbol{I}_K$. Write $\boldsymbol{\Sigma}_t(\boldsymbol{x}_t) = \boldsymbol{U}_t \Lambda_t \boldsymbol{U}_t^\top$ with $\boldsymbol{U}^t$ orthogonal. Set $\boldsymbol{Q} \coloneqq \boldsymbol{U}_t^\top \boldsymbol{V} \in \mathbb{R}^{n \times K}$. Since $\boldsymbol{U}_t$ is orthogonal, $\boldsymbol{Q}^\top \boldsymbol{Q} = \boldsymbol{V}^\top \boldsymbol{U}_t \boldsymbol{U}_t^\top \boldsymbol{V} = \boldsymbol{I}_K$, so the columns of $\boldsymbol{Q}$ are orthonormal.

We have

$$\mathrm{tr}(\boldsymbol{V}^\top \boldsymbol{\Sigma}_t(\boldsymbol{x}_t)\boldsymbol{V}) = \mathrm{tr}(\boldsymbol{V}^\top \boldsymbol{U}_t \Lambda_t \boldsymbol{U}_t^\top \boldsymbol{V}) = \mathrm{tr}(\boldsymbol{Q}^\top \Lambda_t \boldsymbol{Q}) = \sum_{i=1}^n \lambda_i^t \|q_i\|_2^2,$$

where $q_i^\top$ denotes the $i$-th row of $\boldsymbol{Q}$ (so $\|q_i\|_2^2 \geq 0$). Because $\boldsymbol{Q}^\top \boldsymbol{Q} = \boldsymbol{I}_K$,

$$\sum_{i=1}^n \|q_i\|_2^2 = \mathrm{tr}(\boldsymbol{Q}^\top \boldsymbol{Q}) = K, \qquad \text{and} \qquad \|q_i\|_2^2 \leq 1 \text{ for all } i \text{ (each row is a subvector of a unit vector).}$$

Thus the objective is a linear functional of the nonnegative weights $w_i \coloneqq \|q_i\|_2^2$ subject to $\sum_i w_i = K$ and $0 \leq w_i \leq 1$. Since the eigenvalues are ordered $\lambda_1^t \geq \cdots \geq \lambda_n^t$, the sum $\sum_i \lambda_i w_i$ is maximized by assigning $w_i = 1$ for $i = 1, \ldots, K$ and $w_i = 0$ otherwise (rearrangement/greedy argument), which yields

$$\max_{\boldsymbol{V}^\top \boldsymbol{V} = \boldsymbol{I}_K} \mathrm{tr}(\boldsymbol{V}^\top \boldsymbol{\Sigma} \boldsymbol{V}) = \sum_{i=1}^K \lambda_i^t.$$

This maximum is attained by taking $\boldsymbol{Q} = [\boldsymbol{e}_1, \ldots, \boldsymbol{e}_K]$, i.e., $\boldsymbol{V} = \boldsymbol{U}\boldsymbol{Q} = [\boldsymbol{u}_1^t, \ldots, \boldsymbol{u}_K^t]$, the matrix of the top-$K$ eigenvectors. $\square$

## B ADDITIONAL DETAILS

### B.1 COMPUTATIONS OF EIGENVALUES

To compute the leading eigenvalues of the posterior covariance efficiently, we follow the Jacobian-free subspace iteration method introduced by Manor & Michaeli (2024). The algorithm approximates Jacobian–vector products using finite differences of the denoiser output, followed by QR orthogonalization to stabilize directions. Iteratively refining these directions yields approximate posterior

---

**Algorithm 2** Efficient computation of posterior principal components (Manor & Michaeli, 2024)

---

**Require:** $N$ (Number of PCs), $K$ (number of iterations), $\mathrm{D}(\cdot)$ (MSE-optimal denoiser) , $\boldsymbol{y}$ (noisy input), $\sigma_t^2$ (noise variance), $c \ll 1$ (linear approx. constant)

1: Initialize $\{\boldsymbol{v}_0^{(i)}\}_{i=1}^N \sim \mathcal{N}(0, \sigma_t^2 \boldsymbol{I})$
2: **for** $k \leftarrow 1$ **to** $K$ **do**
3:      **for** $i \leftarrow 1$ **to** $N$ **do**
4:          $\boldsymbol{v}_k^{(i)} \leftarrow \dfrac{1}{2c}\Big(\mathrm{D}\big(\boldsymbol{y} + c\,\boldsymbol{v}_{k-1}^{(i)}\big) - \mathrm{D}\big(\boldsymbol{y} - c\,\boldsymbol{v}_{k-1}^{(i)}\big)\Big)$
5:      **end for**
6:      $\boldsymbol{Q}, \boldsymbol{R} \leftarrow \text{QR DECOMPOSITION}\Big(\big[\,\boldsymbol{v}_k^{(1)} \,\cdots\, \boldsymbol{v}_k^{(N)}\,\big]\Big)$
7:      $\big[\,\boldsymbol{v}_k^{(1)} \,\cdots\, \boldsymbol{v}_k^{(N)}\,\big] \leftarrow \boldsymbol{Q}$
8: **end for**
9: $\boldsymbol{v}^{(i)} \leftarrow \boldsymbol{v}_K^{(i)}$
10: $\lambda^{(i)} \leftarrow \dfrac{\sigma_t^2}{2c}\,\big\|\mathrm{D}\big(\boldsymbol{y} + c\,\boldsymbol{v}_{K-1}^{(i)}\big) - \mathrm{D}\big(\boldsymbol{y} - c\,\boldsymbol{v}_{K-1}^{(i)}\big)\big\|$

---

principal components and their corresponding eigenvalues. This approach avoids the costly explicit Jacobian calculation while still capturing the dominant spectral structure of the denoising uncertainty. Algorithm 2 reports the calculations of eigenvalues.

### B.2 ON THE SPD ASSUMPTION FOR LEARNED DENOISERS

Our theoretical analysis relies on properties of the optimal MMSE denoiser, whose Jacobian corresponds to the posterior covariance and is therefore symmetric positive semi-definite (SPD). In practice, however, diffusion models employ learned denoisers that only approximate the true score function and are not guaranteed to satisfy this property exactly. As a result, the Jacobian of a learned denoiser may exhibit deviations from ideal covariance structure, including spurious or weakly negative directions.

This observation provides additional insight into the empirical behavior of different uncertainty-based metrics. Scalarized measures such as MSE or trace-based scores aggregate contributions from all eigen-directions, making them more sensitive to approximation errors in the learned Jacobian. In contrast, EigenScore focuses on the dominant eigenvalues, which are more likely to correspond to meaningful uncertainty directions that persist under model approximation. From this perspective, top-$K$ spectral truncation can be viewed as a robust estimator that better approximates the behavior of the true posterior covariance, even when the learned denoiser deviates from the ideal MMSE solution.

### B.3 PARAMETERS FOR THE REPORTED PERFORMANCE

Our method maintains strong and stable performance even without hyperparameter adjustments, demonstrating that EigenScore is not sensitive to the choice of $T$, $I$, $J$, or $K$ and generalizes well across datasets. The fixed setup uses $k = 3$, $T = \{100, 150, 200, 250\}$, $I = 20$, and all aggregation.

### B.4 BASELINES

We compare our method against several generative baselines for OOD detection, including Improved CD (Du et al., 2021), DoS (Morningstar et al., 2021), TT (Nalisnick et al., 2019b), WAIC (Choi et al., 2018), NLL, IC, DDPM-OOD (Graham et al., 2023), LMD (Liu et al., 2023), DiffPath (Heng et al., 2024). We use the official repositories for each method, along with diffusion models trained under the EDM framework (Karras et al., 2022) and the Glow model (Kingma & Dhariwal, 2018). For the Glow-based baselines (DoS, TT, WAIC), we follow (Morningstar et al., 2021). In DoS, we extract three statistics—the log-likelihood, latent log-probability, and Jacobian log-determinant—and fit Kernel Density Estimators (KDE) on the training data for each, summing across statistics to obtain the final score. For TT and WAIC, we adopt the sample-wise versions from (Morningstar et al., 2021): TT measures the deviation of an individual sample's likelihood from the training-set average, while WAIC is computed from five independently trained models using the mean and variance of their log-likelihoods. For diffusion-based baselines, we compute NLL using the official implementation

from OpenAI's improved diffusion Github repository[1] and derive IC by combining this NLL with PNG compression. Other methods (DDPM-OOD[2], LMD[3], DiffPath[4]) are implemented using their official repositories.

## B.5 DATASET AND MODELS

We follow the official train/test splits when training diffusion models. For validation, we randomly sample 500 in-distribution (InD) and 500 OOD images; for testing, we sample a disjoint set of 500 InD and 500 OOD images. All images are resized to $32 \times 32$. All diffusion models are trained using the EDM framework of Karras et al. (2022). The code can be found here [5].

## B.6 ADDITIONAL EXPERIMENTS

### B.6.1 COMPARISON WITH IMAGENET-1K

We have compared our method using ImageNet dataset as InD and SVHN and Texture dataset as OOD. Table 5 report the results.

Table 5: *ImageNet-1K dataset detection results (AUROC). We evaluate on related datasets, including SVHN and Texture as OOD samples. Note the competitive performance of EigenScore over prior diffusion-based approaches.*

| InD | **ImageNet-1K** vs. | | Avg |
|---|---|---|---|
| OOD | SVHN | Texture | |
| DDPM-OOD | 0.208 | 0.570 | 0.389 |
| LMD | 0.901 | 0.486 | 0.693 |
| DiffPath | 0.753 | 0.699 | 0.726 |
| DiffPathV2 (Abdi et al., 2025) | 0.547 | 0.860 | 0.703 |
| **EigenScore** | 0.755 | 0.709 | 0.732 |

### B.6.2 COMPARISON OF C10 AND C100 WITH LSUN, ISUN, AND TEXTURES

We compare our method using CIFAR-10 and CIFAR-100 as InD datasets and LSUN, iSUN, and Textures as OOD datasets. Table 6 reports the results.

Table 6: *C10 and C200 datasets detection results vs. LSUN, iSUN, and Textures. Note the competitive performance of EigenScore over DiffPath.*

| InD | **C10** vs. | | | **C100** vs. | | | Avg |
|---|---|---|---|---|---|---|---|
| OOD | LSUN | iSUN | Textures | LSUN | iSUN | Textures | |
| DiffPath | 0.618 | 0.911 | 0.780 | 0.609 | 0.869 | 0.761 | 0.758 |
| **EigenScore** | 0.977 | 0.974 | 0.756 | 0.877 | 0.879 | 0.627 | 0.848 |

### B.6.3 ABLATION STUDIES ON PARAMETER c OF THE POWER ITERATION

We perform ablation study on the parameter $c$ used in the power-iteration step when computing the top-1 eigenvalue. We evaluate multiple values of $c$ while keeping all other settings fixed: we use the z-score detection method, mean aggregation across steps, and timesteps $\{100, 150, 200, 250, 300\}$. The experiment is conducted on CIFAR-10 (InD) versus CIFAR-100 (OOD). As reported in Table 7,

---

[1] https://github.com/openai/improved-diffusion
[2] https://github.com/marksgraham/ddpm-ood
[3] https://github.com/zhenzhel/lift_map_detect
[4] https://github.com/clear-nus/diffpath
[5] https://drive.google.com/drive/folders/1-iYwjFhu3am9y3_ZrOirg8CILcsafbhc?usp=drive_link

the results show that EigenScore is stable across a wide range of $c$ values, indicating that the method is not sensitive to this parameter and that our default choice provides strong and reliable performance.

Table 7: *Ablating Parameter c*

| c | 0.005 | 0.01 | 0.05 | 0.1 | 0.5 | 1 |
|---|---|---|---|---|---|---|
| AUROC | 0.846 | 0.844 | 0.850 | 0.847 | 0.866 | 0.875 |

### B.6.4 SENSITIVITY OF EIGENSCORE TO DIFFUSION TIMESTEP

Figure 6 shows the OOD detection performance for the CIFAR-100 vs. CelebA pair across all diffusion timesteps. The results reveal that this pair exhibits peak separability at later timesteps (approximately 400–500), where the posterior covariance inflation between the two datasets becomes most pronounced. Earlier timesteps (100–300), used in the ablation table, do not reach this peak region, explaining the lower performance observed there. This behavior reflects dataset-specific sensitivity to the diffusion trajectory rather than instability, and EigenScore consistently improves once the anisotropic covariance modes dominate.

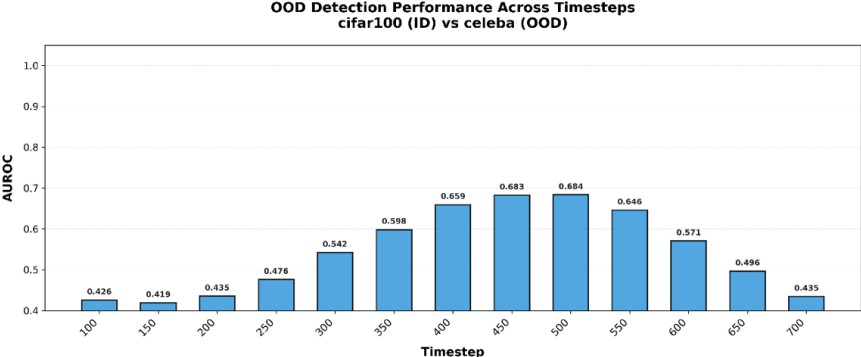

Figure 4: **Sensitivity of EigenScore to diffusion timestep for CIFAR-100 vs. CelebA.** We plot OOD detection performance across all diffusion timesteps. This pair exhibits peak separability at later timesteps (approximately 400–500), where posterior covariance inflation between ID and OOD samples is most pronounced. Earlier timesteps (100–300), used in the ablation study, do not reach this peak region, explaining the lower performance observed there.

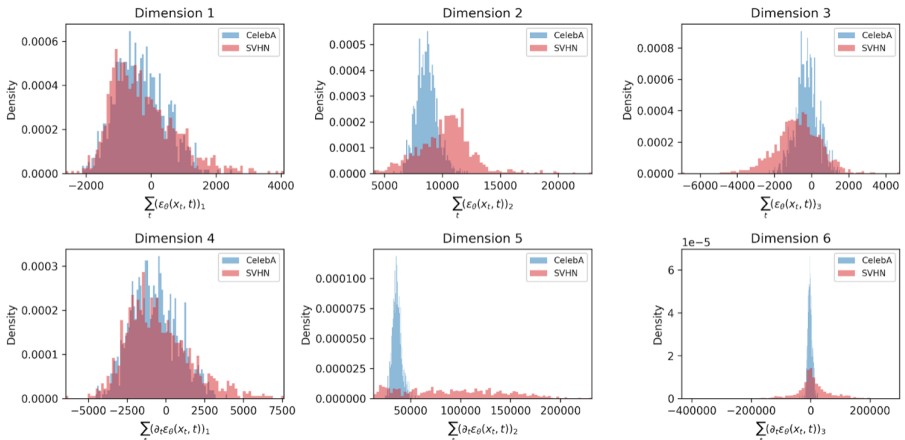

Figure 5: DiffPath feature distributions for SVHN (ID) vs. CelebA (OOD). Each of the six DiffPath dimensions shows substantial overlap between ID and OOD histograms, indicating that the metric provides weak separability for this pair.

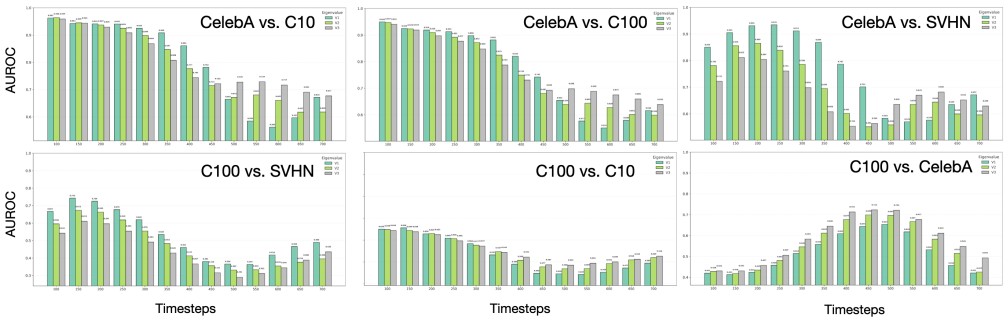

Figure 6: AUROC as a function of diffusion timestep t (100–700) for the largest three eigenvalues (K = 1, 2, 3) across 6 ID/OOD pairs. The plots illustrate how OOD separability evolves along the diffusion trajectory and highlight the timestep regions where additional eigenvalues improve or saturate performance.

### B.6.5 TIME/NFE COMPARISON

To contextualize computational cost, Table 8 reports the number of function evaluations (NFE) and wall-clock time for key baselines and our method per image.

Table 8: *Time/NFE comparisons*

| Method | DiffPath | LMD | DDPM-OOD | EigenScore |
|---|---|---|---|---|
| NFE | 10 | 10000 | 350 | 300 |
| Time (s) | 0.1s | 1.8s | 0.53s | 1.9s |

### B.6.6 HIGH-RESOLUTION OOD DETECTION EXPERIMENTS (256×256)

We evaluate EigenScore on high-resolution and large-scale image datasets. We conduct additional OOD detection experiments using a 256×256 DDPM [6] (Baranchuk et al., 2022). We evaluate OOD

---

[6]https://github.com/yandex-research/ddpm-segmentation

performance on AFHQ (Choi et al., 2020) and Microscopy (CHAMMI) (Chen et al., 2023).Results are reported in Table 9.

Table 9: OOD detection results on high-resolution 256×256 models. In-distribution dataset is FFHQ-256. OOD datasets are AFHQ and Microscopy (CHAMMI).

| Method | AFHQ | Microscopy | Avg |
|---|---|---|---|
| LMD | 0.485 | 0.552 | 0.5185 |
| DiffPath | 0.593 | 0.998 | 0.7955 |
| EigenScore | 0.532 | 0.899 | 0.7155 |

