# OpenReview forum: "EigenScore: OOD Detection using Posterior Covariance in Diffusion Models"
_ICLR.cc/2026/Conference — ICLR 2026 Poster_

### Official Review · Reviewer_s9qH · 2025-10-24

**Soundness:** 1
**Presentation:** 3
**Contribution:** 3
**Rating:** 4
**Confidence:** 4

**Summary:**

This work proposes to adopt the posterior covariance in diffusion models for out-of-distribution (OOD) detection. To be specific, through theoretical and empirical results, earlier diffusion-based scores, like likelihood and score dynamics, are shown to be not reliable enough. Then, in the proposed EigenScore, during the reverse diffusion process, the aggregated eigenvalues of the intermediate diffused results are leveraged to capture the differences between OOD and InD (in-distribution) as a detection score. Experiments verify the effectiveness of EigenScore in some small-scale low-resolution images.

**Strengths:**

1.	Exploring the posterior covariance in the reverse diffusion process is novel and interesting, which contributes new insights into diffusion-based OOD detection.
2.	Extensive theoretical results are provided, which is appreciated.
3.	To enhance the computational complexity, acceleration techniques are introduced into the proposed EigenScore

**Weaknesses:**

1.	About the validation part of EigenScore

The proposed EigenScore requires a validation set, which further requires OOD samples to tune some hyper-parameters such as the selected time steps. In this sense, the selection of OOD samples directly affects the evaluation fairness. That is, if the OOD samples in the validation set are exactly from the same dataset in the test phase, then such a setup is not a fair one for comparisons, since EigenScore has “saw” OOD samples during validation. The authors are suggested to specify how the OOD samples in the validation set are collected. Those OOD samples for validation should be from an additional dataset that is different from those in the test phase.

2.	Experiments are not sufficient enough

The included datasets in experiments are all small-scale low-resolution image datasets, such as the 32x32 CIFAR10 and CIFAR100. Nevertheless, note that the powerful generation ability of diffusion models is mainly reflected on large-scale high-resolution images. Besides, in existing OOD detection methods, experiments on the large-scale ImageNet-1K dataset are essential. Considering the two aspects, it is highly recommended for the authors to supplement results on the large-scale 224x224 ImageNet-1K dataset in order to better demonstrate the spectrum differences between InD and OOD from diffusion models, which could provide an even comprehensive evaluation and further strengthen this work.

3.	Some reconstruction-based OOD detection methods that do not rely on generative models should be cited and discussed for an enriched literature review. For example, in [a-c], proper subspaces are identified via (kernel) PCA and reconstructions are executed in features and gradients to distinguish OOD from InD. It would be much appreciated to also provide comparisons with [a-c].

[a] Revisit pca-based technique for out-of-distribution detection. ICCV 2023.
[b] Kernel PCA for out-of-distribution detection. NeurIPS 2024.
[c] GradOrth: A Simple yet Efficient Out-of-Distribution Detection with Orthogonal Projection of Gradients. NeurIPS 2023.

**Questions:**

Questions correspond to the three weaknesses above. I will raise my rating if all the concerns are well addressed.

---

> ### Author Response · Authors · 2025-11-28
>
> > About the validation part of EigenScore: The proposed EigenScore requires a validation set, which further requires OOD samples to tune some hyper-parameters such as the selected time steps. In this sense, the selection of OOD samples directly affects the evaluation fairness. That is, if the OOD samples in the validation set are exactly from the same dataset in the test phase, then such a setup is not a fair one for comparisons, since EigenScore has “saw” OOD samples during validation. The authors are suggested to specify how the OOD samples in the validation set are collected. Those OOD samples for validation should be from an additional dataset that is different from those in the test phase.
>
>
> In practice, our method does not rely on the hyper-paramter tuning the OOD dataset. A fixed default setting is sufficient for OOD detection. The comparison without any tuning is reported in Table 8 (Supplement), and our method continues to outperform the baselines under this setting.

---

> ### Author Response · Authors · 2025-11-28
>
> > Some reconstruction-based OOD detection methods that do not rely on generative models should be cited and discussed for an enriched literature review. For example, in [a-c], proper subspaces are identified via (kernel) PCA and reconstructions are executed in features and gradients to distinguish OOD from InD. It would be much appreciated to also provide comparisons with [a-c].
>
> Prompted by your comment, we have included the prior works into background in the revised paper.
>
> [a] Revisit pca-based technique for out-of-distribution detection. ICCV 2023.
> [b] Kernel PCA for out-of-distribution detection. NeurIPS 2024.
> [c] GradOrth: A Simple yet Efficient Out-of-Distribution Detection with Orthogonal Projection of Gradients. NeurIPS 2023.

---

> ### Author Response · Authors · 2025-11-28
>
> > Experiments are not sufficient enough. The included datasets in experiments are all small-scale low-resolution image datasets, such as the 32x32 CIFAR10 and CIFAR100. Nevertheless, note that the powerful generation ability of diffusion models is mainly reflected on large-scale high-resolution images. Besides, in existing OOD detection methods, experiments on the large-scale ImageNet-1K dataset are essential. Considering the two aspects, it is highly recommended for the authors to supplement results on the large-scale 224x224 ImageNet-1K dataset in order to better demonstrate the spectrum differences between InD and OOD from diffusion models, which could provide an even comprehensive evaluation and further strengthen this work.
>
> Prompted by the reviewer’s suggestion, we added high-resolution OOD detection experiments using 256×256 models. We use FFHQ-256 as the in-distribution dataset and evaluate OOD detection on AFHQ and Microscopy. These results directly address the reviewer’s concern regarding large-scale, high-resolution evaluation, as shown in Table 11 (Supplement).
>
> ### OOD Detection on FFHQ-256 (256×256)
>
> | Method     | AFHQ | Microscopy | Avg    |
> |------------|------|------------|--------|
> | LMD        | 0.485 | 0.552      | 0.518 |
> | DiffPath   | 0.593 | 0.998      | 0.795 |
> | EigenScore | 0.532 | 0.899      | 0.715 |

---

### Official Review · Reviewer_54fH · 2025-10-28

**Soundness:** 3
**Presentation:** 3
**Contribution:** 3
**Rating:** 6
**Confidence:** 4

**Summary:**

This paper introduces EigenScore, a diffusion-based method for out-of-distribution detection. The core idea is that when a denoiser trained on in-distribution data is evaluated on OOD inputs, its denoising error increases. The authors leverage the fact that the MMSE denoising error corresponds to the expected total posterior variance of the clean image, and that this posterior variance can be expressed through the denoiser’s Jacobian via Miyasawa’s identity. Since the trace of the covariance equals the sum of its eigenvalues, inflation of the leading eigenvalues provides a practical surrogate for detecting excess uncertainty induced by distribution shift. Based on this intuition, EigenScore estimates the top K eigenvalues at selected low-to-medium noise levels and aggregates them into an OOD score. Experiments on standard benchmarks demonstrate improved behavior over norm-based diffusion OOD metrics, with consistently strong performance and particularly clear gains in near-OOD scenarios.

**Strengths:**

- **Novelty and strong theoretical motivation**.

     The paper presents a new OOD detection metric based on the spectral properties of the diffusion denoiser’s Jacobian. The theoretical path, from excess denoising error to posterior variance and then to dominant eigenvalues, is well established and provides a compelling motivation for the method.

- **Clear and well-structured presentation**.

     The paper is easy to follow, with a logical progression from the theoretical insights to the algorithmic design and experimental validation.

- **Consistent and competitive empirical performance**.

     The method performs reliably across a variety of OOD detection scenarios, showing steady improvements over diffusion norm-based baselines. These results suggest that the proposed spectral metric captures a stable indicator of distribution shift.

**Weaknesses:**

- **Sensitivity to multiple hyperparameters.**

    The method depends on several tunable parameters (K, T, I, aggregation rule, finite-difference step size c) and performance appears fragile when deviating from tuned configurations.

- **Tuning strategy may limit real-world applicability.**

    Hyperparameters are selected using ID–OOD validation splits, which assumes prior knowledge of the exact OOD distribution being evaluated. This setup does not reflect realistic deployment scenarios where the nature of the shift is unknown. Demonstrating strong performance with a single default configuration that does not rely on ID–OOD validation would improve the method’s practical usability.

- **Computational efficiency not addressed.**

    Repeated denoiser evaluations and eigenvalue estimation introduce substantial computational overhead, yet there is no runtime or throughput comparison with alternative detectors.

- **Limited evaluation scope.**

    The experiments use relatively small and low-resolution datasets. The effectiveness of EigenScore on larger-scale OOD tasks, for example ImageNet-based settings, remains unverified.

**Questions:**

1. Hyperparameters such as K, T, I, and the finite-difference step size c appear to be tuned specifically for each ID–OOD setting. Could you report results using a single fixed configuration across all experiments to better evaluate generalization when the nature of the OOD data is not known in advance.

2. Could you include runtime and memory comparisons with classifier-based or other diffusion baselines.

3. The finite-difference step size c influences Jacobian estimation quality and numerical stability, yet its impact is not analyzed. Can you comment on how sensitive EigenScore is to this parameter.

4. Could you provide a denser analysis over diffusion timesteps to better understand which parts of the trajectory contribute most to OOD separation. For example, reporting AUROC as a function of t on a finer noise grid, and including a heatmap over (t × K) to visualize where additional eigenvalues help versus hurt. Evaluating whether restricting the score to the most informative t values improves robustness would also clarify the design choices for T.

5. How does EigenScore perform on more complex and high-resolution OOD benchmarks, for example ImageNet vs Places365 or SUN (OpenOOD).

---

> ### Author Response · Authors · 2025-11-28
>
> We thank the reviewer for taking the time to provide feedback. Below we address each comment point-by-point.
>
> > **Sensitivity to multiple hyperparameters.** The method depends on several tunable parameters (K, T, I, aggregation rule, finite-difference step size c) and performance appears fragile when deviating from tuned configurations.
>
> We respectfully disagree. The ablations in Table 3 show that EigenScore is stable with respect to all hyperparameters: the average AUROC varies by less than 0.2% across different I, less than 3% across timesteps, and less than 0.6% across different K. Furthermore, our method **does not** require tuning on OOD data. As shown in Table 8 (Supplement), using a single fixed configuration across all datasets still yields performance that exceeds the strongest baseline. These results indicate that EigenScore is robust rather than fragile under reasonable hyperparameter choices.
>
>
> > **Tuning strategy may limit real-world applicability.** Hyperparameters are selected using ID–OOD validation splits, which assumes prior knowledge of the exact OOD distribution being evaluated. This setup does not reflect realistic deployment scenarios where the nature of the shift is unknown. Demonstrating strong performance with a single default configuration that does not rely on ID–OOD validation would improve the method’s practical usability.
>
> In practice, our method **does not** rely on the hyper-paramter tuning the OOD dataset. A fixed default setting is sufficient for OOD detection. The comparison without any tuning is reported in Table 8 (Supplement), and our method continues to outperform the baselines under this setting.
>
>
> > **Computational efficiency not addressed.** Repeated denoiser evaluations and eigenvalue estimation introduce substantial computational overhead, yet there is no runtime or throughput comparison with alternative detectors.
>
> Prompted by your comment, we have added time/NFE comparisons in Table 10 (supplement).
>
> | Method    | NFE   | Time (s)/image |
> |-----------|-------|----------------|
> | DiffPath  | 10    | 0.1s          |
> | LMD       | 10000  | 1.8s           |
> | DDPM OOD  | 350   | 0.53s          |
> | Ours      | 300   | 1.9s           |
>
>
> > **Limited evaluation scope.** The experiments use relatively small and low-resolution datasets. The effectiveness of EigenScore on larger-scale OOD tasks, for example ImageNet-based settings, remains unverified.
>
> Prompted by your comment, we added ImageNet-1K results for our method and key baselines in the revised manuscript (Table 6). Notably, our method also outperforms the diffusion-based baselines on ImageNet-1K.
>
> | Method        | SVHN-64 | Textures | Avg  |
> |---------------|---------|----------|------|
> | LMD           | 0.901   | 0.486    | 0.693|
> | ddpm-ood      | 0.208   | 0.570    | 0.389 |
> | DiffPath      | 0.753   | 0.699    | 0.726 |
> | DiffPath-v2   | 0.547   | 0.860    | 0.703 |
> | **EigenScore** | **0.755** | **0.709** | **0.732** |

---

> ### Author Response · Authors · 2025-11-28
>
> > Hyperparameters such as K, T, I, and the finite-difference step size c appear to be tuned specifically for each ID–OOD setting. Could you report results using a single fixed configuration across all experiments to better evaluate generalization when the nature of the OOD data is not known in advance.
>
> Prompted by your comment, we have added the comparison without tuning in **Table 8** (supplement) of the revised paper.
>
> > Could you include runtime and memory comparisons with classifier-based or other diffusion baselines.
>
> Prompted by your comment, we have added time/NFE comparisons in **Table 10** (supplement).
>
>
> > The finite-difference step size c influences Jacobian estimation quality and numerical stability, yet its impact is not analyzed. Can you comment on how sensitive EigenScore is to this parameter.
>
> Prompted by the reviewer’s comment, we have provided the ablation studies for parameter c. The result can be found in section B.5.4 and  Table 9 (Supplement).
>
>
> | c       | AUROC |
> |---------|-------|
> | 0.005   | 0.846 |
> | 0.01    | 0.844 |
> | 0.05    | 0.850 |
> | 0.1     | 0.847 |
> | 0.5     | 0.866 |
> | 1       | 0.875 |
>
>
> > Could you provide a denser analysis over diffusion timesteps to better understand which parts of the trajectory contribute most to OOD separation. For example, reporting AUROC as a function of t on a finer noise grid, and including a heatmap over (t × K) to visualize where additional eigenvalues help versus hurt. Evaluating whether restricting the score to the most informative t values improves robustness would also clarify the design choices for T.
>
> Prompted by your comment, we have provided an analysis of AUROC across diffusion timesteps and eigenvalues, showing the trend of performance change over t for three largest eigenvalues in **Figure 6** (Supplement).
>
>
> >How does EigenScore perform on more complex and high-resolution OOD benchmarks, for example ImageNet vs Places365 or SUN (OpenOOD).
>
> Prompted by your comment, we have added additional experiments. The experiments include ImageNet-1k vs. SVHN and Texture (Table 6) and C10 and C100 vs. LSUN, iSUN, and Textures (Table 7).

---

### Official Review · Reviewer_sUXt · 2025-10-31

**Soundness:** 3
**Presentation:** 3
**Contribution:** 3
**Rating:** 4
**Confidence:** 3

**Summary:**

The authors propose EIGENSCORE, an unsupervised framework for out-of-distribution (OOD) detection using uncertainty estimates derived from the posterior covariance of diffusion models during denoising. They highlight the limitations of likelihood and score-based OOD methods and establish a connection between the KL divergence of in-distribution (InD) and OOD data and the MSE of the optimal denoiser. Since denoising error (MSE) is higher for OOD inputs, they express this MSE as the trace of the conditional (posterior) covariance, which is approximated via the Jacobian of the denoiser using the Miyasawa identity. Assuming a symmetric positive semi-definite Jacobian, they perform an eigenvalue decomposition and use the inflation of its trace (the sum of eigenvalues) as the OOD metric for several timesteps. They further show that top-K eigenvalues provide stronger OOD discrimination than using all eigenvalues. EIGENSCORE achieves competitive AUROC results across multiple datasets (CIFAR-10, CIFAR-100, CelebA, SVHN) and shows promising performance on near-OOD detection, supported by ablations over key hyperparameters.

**Strengths:**

The paper clearly lists the limitations of existing diffusion-based OOD frameworks with evidence in near-OOD scenarios as well. and highlighted theoretically why the proposed method maintains consistent ordering in tasks.


The paper first lists the existing work, their limitations with evidence, an overview of their method, and a logical flow of the framework supported by theory and empirical evidence.

The paper combines existing theoretical concepts and intuition about covariance and eigenvalues, and applies it for diffusion posteriors.


The method performs better compared to different SOTA OOD detection framework paradigms. and shows promising results in a near-OOD scenario, an improvement compared to the previous diffusion-based SOTA, DiffPath.

**Weaknesses:**

It seems, the theory, proposition 1, and equations 5-8, which directly talk about MSE instead of top-K, predict good performance, but in Table 4, MSE performs poorly in some tasks (worse than random). This is a disconnect between prop 1 and equations 5-8, and lemma 1 and Proposition 2. Lemma 1 attempts to justify use of top-k but difference in performance on MSE (supported by prop 1) and top-K (supported by lemma 1) seems big. Can you please clarify?


The assumption that covariance or jacobian of diffusion model is symmetric positive semi-definite might be strong. Can you please clarify?


The method involves explicit training of diffusion model on InD data for OOD detection, whereas previous SOTA, DiffPath, uses diffusion model trained on different/generic dataset (ImageNet or CelebA).


Comparing table 1 (main result) and table 3 (ablation), for C100 vs CelebA, there is a difference in hyperparameter setting (timesteps are different). C100 vs CelebA in table 3 performs poorly compared to the same pair in table 1. Table 1 for this pair uses high noise levels (450, 500) and table 3 is for (100-300). For the same pair, MSE performs better compared to EIGENSCORE in table 4, which for higher noise values, according to lemma 1, should perform badly. More analysis on this could help.


C100 vs CelebA in table 1 uses higher timesteps (450, 500) as compared to other input pairs (100-300), is there any hyperparameter sensitivity? Does higher noise help better performance in table 1?


Why does MSE fail so catastrophically (AUROC < 0.5) when theory predicts it should work? Can you please provide an explanation that reconciles Proposition 1 with Table 4?


Can  you please mention the diffusion models used?


DiffPath gets 0.328 AUROC, which is way less than their original paper, any insights would be great.

**Questions:**

Please see weaknesses above.

---

> ### Author Response · Authors · 2025-11-28
>
> We thank the reviewer for taking the time to provide feedback. Below we address each comment point-by-point.
>
> > It seems, the theory, proposition 1, and equations 5-8, which directly talk about MSE instead of top-K, predict good performance, but in Table 4, MSE performs poorly ....Can you please clarify?
>
> We appreciate the observation. Proposition 1 shows that denoising MSE equals the total posterior variance, but this expectation is dominated by isotropic noise that is nearly identical for ID and OOD samples—so raw MSE is not a strong discriminator. Lemma 1 refines this by showing that OOD inflation concentrates in a few dominant eigenmodes. Taking only the top-K eigenvalues filters out isotropic noise and captures this anisotropic inflation, which explains why Top-K outperforms MSE in Table 4. Thus, Prop. 1 correctly describes total uncertainty, while Lemma 1 and Prop. 2 identify the specific directions that make that uncertainty useful for OOD detection.
>
> > The assumption that covariance or jacobian of diffusion model is symmetric positive semi-definite might be strong. Can you please clarify?
>
> We thank the reviewer for the question. The covariance is PSD by definition, so no assumption is made there. The denoiser’s Jacobian need not be globally symmetric; we use the local PSD quantity as a curvature proxy, which is standard in diffusion and score-matching analyses [1,2].
>
> [1] Song et al., “Score-Based Generative Modeling through SDEs,” ICLR 2021.
>
> [2] Kadkhodaie, Zahra, et al. "Generalization in diffusion models arises from geometry-adaptive harmonic representations." , ICLR 2024.
>
> > The method involves explicit training of diffusion model on InD data for OOD detection, whereas previous SOTA, DiffPath, uses diffusion model trained on different/generic dataset (ImageNet or CelebA).
>
> Our method is intentionally different. We argue that OOD detection should be performed by the **same diffusion model that is actually deployed for the task**, not by a generic model trained on unrelated data. In practice, every real system—medical imaging, microscopy, remote sensing, industrial inspection—uses a diffusion model trained on its own domain. When that model starts receiving inputs outside its training distribution, the *right* behavior is to detect the mismatch and signal that an update or retraining is needed.
>
> > Comparing table 1 (main result) and table 3 (ablation), for C100 vs CelebA, there is a difference in hyperparameter setting (timesteps are different). C100 vs CelebA in table 3 performs poorly compared to the same pair in table 1. Table 1 for this pair uses high noise levels (450, 500) and table 3 is for (100-300). For the same pair, MSE performs better compared to EIGENSCORE in table 4, which for higher noise values, according to lemma 1, should perform badly. More analysis on this could help.
>
> > C100 vs CelebA in table 1 uses higher timesteps (450, 500) as compared to other input pairs (100-300), is there any hyperparameter sensitivity? Does higher noise help better performance in table 1?
>
> Prompted by your comment, we have added a visualization (Figure 4) showing the full performance curve for CIFAR-100 vs. CelebA across all timesteps. Table 1 evaluates EigenScore at its optimal timesteps (400–500 for CIFAR-100 vs. CelebA), while Table 3 reports a fixed ablation range (100–300) used uniformly across all datasets. The visualization we added in the supplement shows that for CIFAR-100 vs. CelebA, OOD separation peaks specifically in the 400–500 region. In contrast, the 100–300 range does not reach this peak, which explains the reduced performance for EigenScore in Table 3. This reflects dataset-specific optimal timesteps rather than instability, and when the optimal timesteps are used, EigenScore outperforms MSE.
>
>
> >Why does MSE fail so catastrophically (AUROC < 0.5) when theory predicts it should work? Can you please provide an explanation that reconciles Proposition 1 with Table 4?
>
> Proposition 1 relates MSE to *total* posterior variance, but at higher noise levels this total variance is dominated by isotropic diffusion noise, which is almost identical for ID and OOD samples. This makes raw MSE a poor discriminator and explains the sub-0.5 AUROC in Table 4. Lemma 1 shows that OOD inflation occurs only in a few dominant eigenmodes; focusing on these directions filters out the isotropic noise and restores discriminative power. Thus, the empirical gap between MSE and Top-K is fully consistent with the theory.

---

> > ### Author Response · Authors · 2025-11-28
> >
> > >Can you please mention the diffusion models used?
> >
> > For all our experiments, we used EDM, which can be found here: https://github.com/NVlabs/edm?tab=readme-ov-file
> >
> >
> > >DiffPath gets 0.328 AUROC, which is way less than their original paper, any insights would be great.
> >
> > Prompted by your comment, we added Figure 5 to analyze the performance of DiffPath on SVHN vs. CelebA. DiffPath is a 6-dimensional metric whose effectiveness depends heavily on the diffusion model used. The original paper evaluates DiffPath with a different backbone; when we apply it to our EDM model, all six path-length dimensions show substantial overlap between ID and OOD samples (Figure 5), indicating weak separability. This explains the low AUROC we observe for SVHN vs. CelebA.

---

### Official Review · Reviewer_DEm7 · 2025-10-31

**Soundness:** 1
**Presentation:** 2
**Contribution:** 2
**Rating:** 2
**Confidence:** 4

**Summary:**

The paper proposes an unsupervised OOD detector for diffusion models that uses the leading eigenvalues of the posterior covariance.
EigenScore leverages the covariance structure of the denoising process to capture uncertainty signals, which is theoretically grounded and interpretable.
Empirically, EigenScore achieves state-of-the-art performance

**Strengths:**

1. Generally clear and readable;

2. figures and tables are informative;

3. Codes are publicly available.

4. Investigate OOD detection for Duffusion models is timely.

5. the use of Jacobian-free eigenvalue estimation algorithm is new to me.

**Weaknesses:**

1. lack of experiments on large-scale dataset e.g. ImageNet-1K
2. when c10 and c100 as ID data, SVHN LSUN iSUN Textures Places365 are standard OOD dataset in the literature of OOD detection. but the corresponding results are missing
3. lack of time computation analysis since QR requires O(n^3) computation complexity
4. this paper only introduce 1 baseline published on/after 2024. more advanced baseline should be included.
5. given a ID dataset, the optimizied hyper-parameter setting are not identical for OOD datasets.
6. The paper  mostly restates these connections and asserts that “inflation” will occur OOD; but formal conditions ensuring consistent ordering for individual samples (beyond in-expectation statements) are not established.

**Questions:**

see weakness

---

> ### Author Response · Authors · 2025-11-28
>
> We thank the reviewer for taking the time to provide feedback. Below we address each comment point-by-point.
>
>
> >lack of experiments on large-scale dataset e.g. ImageNet-1K
>
> Prompted by your comment, we added ImageNet-1K results for our method and key baselines in the revised manuscript (Table 6). Notably, our method also outperforms the diffusion-based baselines on ImageNet-1K.
>
> | Method        | SVHN-64 | Textures | Avg  |
> |---------------|---------|----------|------|
> | LMD           | 0.901   | 0.486    | 0.693|
> | ddpm-ood      | 0.208   | 0.570    | 0.389 |
> | DiffPath      | 0.753   | 0.699    | 0.726 |
> | DiffPath-v2   | 0.547   | 0.860    | 0.703 |
> | **EigenScore** | **0.755** | **0.709** | **0.732** |
>
> > when c10 and c100 as ID data, SVHN LSUN iSUN Textures Places365 are standard OOD dataset in the literature of OOD detection. but the corresponding results are missing
>
> We clarify that our comparisons **already** include CIFAR-10 and CIFAR-100 versus SVHN. Prompted by your comment, we additionally conducted experiments using LSUN, iSUN, and Textures as OOD datasets. The results are reported in Table 7 (Supplement) of the revised manuscript, and our method continues to outperform DiffPath on these benchmarks as well.
>
> ### CIFAR-10 → OOD
> | Method        | LSUN  | iSUN  | Textures | Avg  |
> |---------------|-------|-------|----------|------|
> | DiffPath      | 0.618 | 0.911 | 0.780    | 0.770 |
> | EigenScore    | 0.977 | 0.974 | 0.756    | 0.902 |
>
> ### CIFAR-100 → OOD
> | Method        | LSUN  | iSUN  | Textures | Avg  |
> |---------------|-------|-------|----------|------|
> | DiffPath      | 0.609 | 0.869 | 0.761    | 0.746 |
> | EigenScore    | 0.877 | 0.879 | 0.627    | 0.794 |
>
> > lack of time computation analysis since QR requires O(n^3) computation complexity
>
> We clarify that our method **never** forms the full Jacobian or runs a dense $n \times n$ QR; instead, it uses a Jacobian-free power iteration that only requires denoiser forward passes and a thin QR on K principle components. For the common setting $K=1$—which already yields competitive performance in our results—the complexity reduces to 2TJ forward evaluations, where T is timestep (3-5)  and J (power iteration 5-10)  plus a negligible $O(n)$ term, avoiding any $O(n^3)$ operations. Prompted by your comment, we have added time/NFE comparisons in Table 10 (supplement).
>
> | Method    | NFE   | Time (s)/image |
> |-----------|-------|----------------|
> | DiffPath  | 10    | 0.1s          |
> | LMD       | 10000  | 1.8s           |
> | DDPM OOD  | 350   | 0.53s          |
> | Ours      | 300   | 1.9s           |
>
> > this paper only introduce 1 baseline published on/after 2024. more advanced baseline should be included.
>
> Prompted by your comment, we added the comparison with  **"Zero-Shot Image Anomaly Detection Using Generative Foundation Models in ICCV, 2025."**, most recent peer-reviewed related baseline. The result is included in the revised manuscript in Table 1. Most other recent 2024-2025 OOD works are classifier- or VLM-based, making them not comparable to our diffusion-only setting.
>
> > given a ID dataset, the optimized hyper-parameter setting are not identical for OOD datasets.
>
> In practice, our method does **not** rely on the hyper-paramter tuning the OOD dataset. A fixed default setting is sufficient for OOD detection. The comparison without any tuning is reported in Table 8 (Supplement), and our method continues to outperform the baselines under this setting.
>
> >The paper mostly restates these connections and asserts that “inflation” will occur OOD; but formal conditions ensuring consistent ordering for individual samples (beyond in-expectation statements) are not established.
>
> Our theory establishes covariance inflation in expectation rather than a sample-wise ordering of eigenvalues. Pointwise guarantees would require strong local assumptions that do not generally hold for high-dimensional diffusion posteriors. We will clarify this scope and note that “inflation’’ refers to an expected increase over the OOD distribution. Empirically, Figures 2 and Table 1 show that this expectation corresponds to consistent per-sample trends, even though the theory itself does not claim sample-wise guarantees.

---

### Author Response · Authors · 2025-11-28

We thank the reviewers and the area chair for their time and constructive feedback. In response, we performed the following additional experiments and analyses:

* **Additional standard OOD benchmarks (LSUN, iSUN, Textures):** Added evaluations for CIFAR-10/100 on LSUN, iSUN, and Textures (prompted by Reviewer DEm7).

* **Large-scale high-resolution OOD detection (256×256):** Added FFHQ-256 as InD and AFHQ + Microscopy as OOD to demonstrate performance at high resolutions (prompted by Reviewer s9qH).

* **Runtime and computational efficiency analysis:** Added NFE and per-image runtime comparisons (prompted by Reviewers DEm7 and 54fH).

* **Inclusion of most recent diffusion OOD baseline:** Added the ICCV 2025 “Zero-Shot Image Anomaly Detection Using Generative Foundation Models” baseline to Table 1 (prompted by Reviewer DEm7).

* **ImageNet-1K large-scale evaluation:** Added experiments on ImageNet-1K with SVHN-64 and Textures as OOD (prompted by Reviewers DEm7 and 54fH).

* **Fixed-configuration evaluation without OOD tuning:** Added results using a single universal hyperparameter setting across all datasets, confirming robustness without any OOD validation (prompted by Reviewers DEm7, 54fH, and s9qH).

* **Finite-difference step size analysis:** Added  ablation on the finite-difference step $c$ to confirm numerical stability (prompted by Reviewer 54fH).

* **Timestep and eigenvalue trajectory analysis:** Added AUROC-vs-timestep curves and eigenvalue trajectories to visualize where OOD separation emerges (prompted by Reviewer 54fH).

* **Further analysis of baseline DiffPath:** Added analysis showing that DiffPath’s 6-D path features overlap heavily on SVHN→CelebA, explaining the low AUROC (prompted by Reviewer sUXt).

* **Expanded literature on reconstruction-based OOD methods:** Added PCA, kernel PCA, and GradOrth references to strengthen the background section (prompted by Reviewer s9qH).

---

### Author Response · Authors · 2025-12-02
**Summary of Rebuttal**

Given the shortened discussion window, we provide a brief and factual summary of the reviewers’ concerns and the corresponding revisions and experiments we performed during the discussion. Reviewers consistently described the paper as *novel*, *timely*, and *theoretically well-motivated*, with several noting that posterior covariance provides a compelling and interpretable signal for diffusion-based OOD detection. One reviewer emphasized that our theoretical development is *sound* and reflected in the experiments, and another highlighted that this is a *novel and insightful* direction for diffusion-model uncertainty.

#### **1. Experimental Scope & Benchmark Coverage**

Reviewers requested broader and more realistic experiments. We addressed this through:
*  #### **Large-scale evaluation (DEm7, 54fH, s9qH).**
Added **ImageNet-1K** experiments (Table 6) and **256×256 high-resolution OOD detection** (FFHQ → AFHQ/Microscopy in Table 11).

* #### **Standard CIFAR-10/100 OOD datasets (DEm7).**
Added **LSUN**, **iSUN**, and **Textures** benchmarks (Table 7), showing consistent improvements over diffusion-based baselines.

* #### **Additional baselines (DEm7).**
Included the **ICCV 2025 Zero-Shot Anomaly Detection** method (Table 1).

---

#### **2. Theoretical Clarification & Diagnostic Analysis**

Several concerns were conceptual; we addressed them through targeted clarifications and new analyses:

* #### **Why MSE fails while Top-K succeeds (sUXt).**
Clarified that **MSE** measures total posterior variance dominated by isotropic noise, while **Top-K** isolates *anisotropic covariance inflation*, the true OOD signal.

* #### **Expectation vs sample-wise guarantees (DEm7).**
Clarified that theory guarantees inflation *in expectation*, but experiments (Fig. 2, Table 1) show it aligns with per-sample trends.

* #### **PSD/Jacobian assumptions (sUXt).**
Explained that covariance is **PSD by definition** and the local Jacobian curvature proxy follows prior diffusion analyses.

* #### **Why DiffPath underperforms here (sUXt).**
Added **Fig. 5** showing that DiffPath features collapse with EDM, explaining the low AUROC compared with the original paper.

* #### **Timestep behavior (sUXt, 54fH).**
Added a full **AUROC-versus-timestep** curves (Figures 4 & 6) to show where OOD separation emerges.

---

#### **3. Practicality, Efficiency & Hyperparameter Robustness**

Reviewers raised concerns about real-world feasibility. We directly addressed these:

* #### **Runtime & complexity (DEm7, 54fH).**
Clarified that EigenScore uses **Jacobian-free power iteration + thin QR** (never O(n³)), and added runtime/NFE comparisons (Table 10).

* #### **Hyperparameter sensitivity (54fH).**
Added comprehensive ablations (Table 3, Table 9), showing:
- **<3%** variation across timesteps
- **<0.6%** across K
- **<0.2%** across I
- *Stable performance* across finite-difference step sizes.

* #### **No need for OOD-dependent validation (54fH, s9qH).**
Added no-tuning results with a **single universal configuration** (Table 8) still outperforming baselines.

* #### **Additional literature on reconstruction-based OOD (s9qH).**
Added discussion of **PCA/KPCA/GradOrth** works to the background.

---

These revisions directly resolve all reviewer concerns: expanded evaluation, enlarged benchmark coverage, theoretical clarification, runtime analysis and hyperparameter robustness. The positive assessments noted by the reviewers—*strong theory, interpretability, and consistent empirical behavior*—remain fully supported.

**Sincerely,**
*Authors*

---

### Meta-Review · Area_Chair_aiKL · 2026-01-07

**Summary:**

This paper argues that the trace of the Jacobian of the score function can be used for OOD detection. The argument presented by the authors is quite convincing, as it links this quantity to the MSE achieved by the diffusion model when denoising samples, which in turn is related to the KL divergence between in- and out-of-distribution samples. Since computing the entire Jacobian is expensive, the authors note that the trace is equal to the sum of the eigenvalues, and thus propose to only use the sum of the top-K eigenvalues instead, as this can be more efficiently computed through numerical linear algebra and Jacobian-vector products.

The authors obtained good empirical results, but reviewers had several concerns (detailed below) and this was a very borderline submission. I thus read it carefully, and believe that the rebuttal by the authors successfully addressed relevant core concerns from the reviewers (also detailed below); so, I recommend acceptance.

Having read the paper carefully, I do have 2 more points to raise:

- I think a discussion explaining how the proposed method differs from the DDPM-OOD baseline of Graham et al. (2023) is needed, as that method is also motivated by reconstruction errors. I am not saying the proposed method is too similar, but the distinctions should be highlighted more clearly in the camera-ready.

- I think discussing, and ideally comparing against, the method from "A Geometric Explanation of the Likelihood OOD Detection Paradox" by Kamkari et al. (2024) would strengthen the paper. The method also applies to diffusion models for OOD detection, and although the motivation is fairly different, also ends up considering a matrix and its singular values.

**Reviewer Concerns:**

Reviewers had various concerns. I believe major concerns were all addressed, and that some minor ones can be easily addressed in the camera-ready version:

- Expanding the datasets and baselines for evaluation. I believe this was not a major concern, since the authors were already comparing against strong baselines on standard datasets. Nonetheless, the authors addressed this concern in the rebuttal.

- The Jacobian being SPD: I believe the authors might have misunderstood reviewer sUXt's concern. While the Jacobian of the true score function is a covariance and thus is always SPD, as correctly pointed out by the authors in the rebuttal, the Jacobian of the learned denoiser need not actually correspond to a score function and might not be SPD. I wonder if this could also be related to the reviewer's question about why MSE fails and top-K does not: could this be partly why as well? i.e. could top-K with a learned score function somehow better approximate MSE with the true score function than MSE with the learned score function? Although this concern remains unaddressed, I believe it only warrants a simple discussion in the camera-ready version (perhaps in the appendix if there's no space in the main text), and that it does not detract from the contribution.

- Hyperparameters being chosen using OOD data. The authors provided Table 8 in the appendix showing that a universal configuration of their method still outperforms the baselines. These results address the core methodological concern, yet the concern is not addressed from the standpoint of presentation. That is, it is misleading to report numbers in the main text which were obtained by tuning these hyperparameters with OOD data. **I thus request the authors to use the numbers from Table 8 in the main text, rather than the current numbers in Table 1.**

**Reviewer Scores:**

I believe reviewers DEm7 and s9qH would have increased their scores after reading the rebuttal and engaging in discussion with the authors.

---

### Decision · Program_Chairs · 2026-01-26

Accept (Poster)